# Magnetic control of self-assembly and disassembly in organic materials

You-jin Jung[1,3], Hyoseok Kim[1,3], Hae-Kap Cheong[2] & Yong-beom Lim ®[1] ✉

Because organic molecules and materials are generally insensitive or weakly sensitive to magnetic fields, a certain means to enhance their magnetic responsiveness needs to be exploited. Here we show a strategy to amplify the magnetic responsiveness of self-assembled peptide nanostructures by synergistically combining the concepts of perfect α-helix and rod-coil supramolecular building blocks. Firstly, we develop a monomeric, nonpolar, and perfect α-helix (MNP-helix). Then, we employ the MNP-helix as the rod block of rod-coil amphiphiles (rod-coils) because rod-coils are well-suited for fabricating responsive assemblies. We show that the self-assembly processes of the designed rod-coils and disassembly of rod-coil/DNA complexes can be controlled in a magnetically responsive manner using the relatively weak magnetic field provided by the ordinary neodymium magnet [0.07 ~ 0.25 Tesla (T)]. These results demonstrate that magnetically responsive organic assemblies usable under practical conditions can be realized by using rod-coil supramolecular building blocks containing constructively organized diamagnetic moieties.

Magnetically responsive materials are highly valuable for biological and medical uses because the magnetic field is noninvasive, radiation-free, and safe for humans. A wide variety of stimuli-responsive self-assembled materials have been developed to date[1]. Magnetically responsive self-assembled materials are one of them; however, most of these materials are based on ferromagnetic/paramagnetic inorganic materials or the composites of inorganic and organic materials[2,3]. Although magnetically responsive self-assembled materials based solely on organic materials do exist, a limited number of cases have been reported[4,5]. DNA and some proteins have also been shown to be magnetically responsive; however, with very few exceptions, an extremely strong magnetic field is typically required to influence the behaviours of such biomolecules[6–11]. Consequently, in our opinion, the field of magnetically responsive organic materials could not gain widespread attention, which in large part is due to the difficulty of developing organic materials responsive to the magnetic fields with high sensitivity.

All materials including organic materials, display diamagnetism because an external magnetic field will change the movement of electrons and create an internal magnetic field in the opposite direction[12]. However, the diamagnetic force is usually very weak for organic molecules and materials. Because magnetic susceptibility is additive, one possible way of increasing magnetic responsiveness is to place multiple magnetically responsive moieties within a single molecule in a favourable orientation. The α-helix, one of the most common secondary structures of proteins, provides a nice example of constructive addition in diamagnetism. Similar to aromatic rings, the resonance stabilized peptide bond has a meaningful value of diamagnetic anisotropy due to its partial double bond character (Fig. 1a)[13,14]. Because the multiple peptide bonds are oriented parallel to the helical axis, the overall diamagnetism of the α-helix can be amplified relative to that of a single peptide bond (Fig. 1b). The α-helix can align parallel to the magnetic field because it is a rod-shaped molecule with its long helix axis less susceptible to magnetic field[10–12]. Despite this, the magnetic anisotropy values of α-helices are still too small to use α-helices in developing magnetically responsive self-assembled organic materials sensitive enough for practical uses. Thus, an additional mechanism of increasing magnetic responsiveness is required if one

[1]Department of Materials Science & Engineering, Yonsei University, 50 Yonsei-ro, Seoul 03722, Republic of Korea. [2]Division of Magnetic Resonance, Korea Basic Science Institute, Ochang 28119, Republic of Korea. [3]These authors contributed equally: You-jin Jung, Hyoseok Kim. ✉e-mail: yblim@yonsei.ac.kr

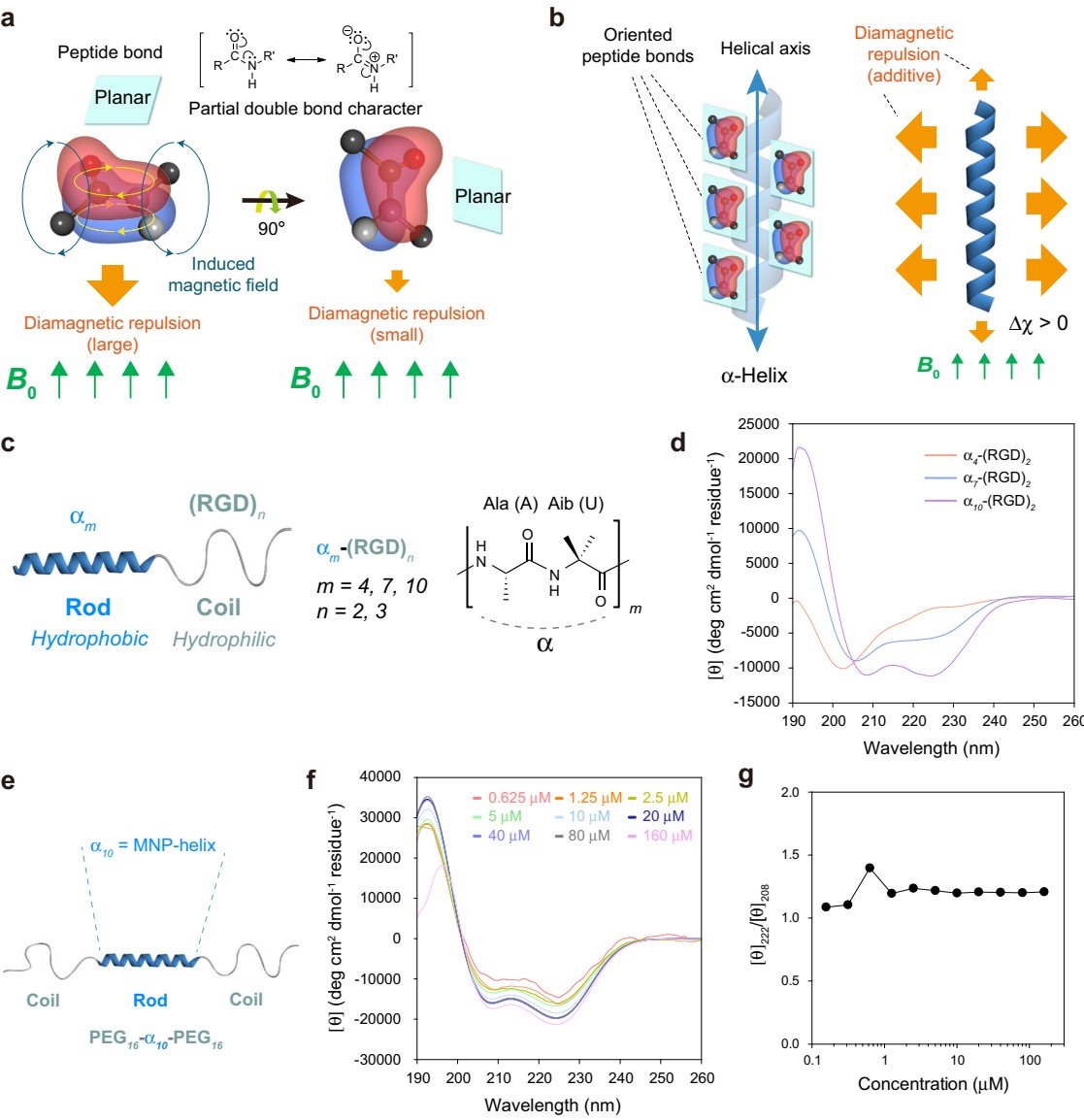

**Fig. 1 | Development of an MNP-helix containing a minimal number of amino acids. a** Diamagnetic anisotropy of the peptide bond. $B_0$: external magnetic field (MF). **b** Difference in diamagnetic susceptibility between the orientations parallel and perpendicular to the helical axis. **c** Design of self-assembling rod-coils for the development of the MNP-helix. **d** Circular dichroism (CD) spectra of $\alpha_m$-(RGD)$_2$ rod-coils. **e** Design of a monomeric coil-rod-coil (PEG$_{16}$-$\alpha_{10}$-PEG$_{16}$). MNP-helix = monomeric & nonpolar & perfect α-helix. **f** CD spectra of PEG$_{16}$-$\alpha_{10}$-PEG$_{16}$ over a wide range of concentrations. **g** Concentration independence of helicity for PEG$_{16}$-$\alpha_{10}$-PEG$_{16}$.

would like to develop magnetically responsive self-assembled organic materials.

Rod-coil block molecules (rod-coils), as supramolecular building blocks, have shown great opportunities in constructing elaborate self-assembled nanostructures[15–17]. For self-assembly in aqueous solutions, rod-coil amphiphiles consist of a rigid, rod-like hydrophobic block and a flexible, coil-like hydrophilic block. During the self-assembly processes, rod-coils have advantages over conventional coil-coils in forming well-defined, dynamic, and responsive assemblies because the rod block imparts orientational organization, and the entropic penalty associated with chain stretching in coil-coils can be reduced in rod-coils. Although α-helices have rod-like characteristics, most of them cannot maintain their helical structures in the monomeric state. In proteins, most α-helices can be stabilized only when they make non-covalent interactions with other residues of proteins[18,19]. Single α-helices found in proteins and some of designed peptides are highly helical in the monomeric state[20,21]; however, these helices cannot be

used as the hydrophobic rod blocks of rod-coils because all of them are highly charged and hydrophilic.

Here, by exploring the synergy of the α-helix and the rod-coil, we developed organic molecular assemblies responsive to the magnetic field produced by common permanent magnets. To this end, the first step was the development of a hydrophobic α-helix that can maintain its almost perfectly helical conformation in the monomeric state (monomeric & nonpolar & perfect α-helix; MNP-helix). By using the MNP-helix as the rod block of rod-coil amphiphiles, we were able to control the self-assembly and disassembly processes under a relatively weak magnetic field (0.07 ~ 0.25 T). To show the magnetically responsive disassembly and the application potential of MNP-helix rod-coils, we designed a positively charged rod-coil that can neutralize and subsequently package negatively charged DNA at high information density in the form of an artificial chromosome. The packaged DNA could be released by the disassembly of the artificial chromosome induced by common permanent magnets. We expect that various

magnetically responsive assemblies can be further developed based on the concept described in this study.

## Results

### Development of an MNP-helix rod

α-Helices are more likely to be stabilized as the chain length increases by the effects of helix nucleation and propagation[18,22]; however, synthetic difficulty associated with excessively long peptides would hamper the process of rod-coil building block development. Thus, we intended to develop the smallest possible MNP-helix. Alanine (Ala or A) has the highest α-helical propensity among standard amino acids[23]; however, its hydrophobicity is the weakest of all hydrophobic amino acids. α-Aminoisobutyric acid (Aib or U) is a strong helix inducer[24] and is more hydrophobic than Ala due to the presence of an additional methyl group. Homopolymer of Aib was excluded from the design because multiple Aib repeats induced the formation of $3_{10}$ helices rather than α-helices. Thus, we anticipated that a certain combination of Ala and Aib in peptides might produce the MNP-helix with an appropriate hydrophobicity for self-assembly. We considered the possibility of using the repeats of Ala-Aib or Ala-Aib-Aib. Because we found that the synthetic yield dropped significantly for a contiguous stretch of Aib, we decided to employ the Ala-Aib repeat in designing MNP-helix. Because it is not possible to investigate the behaviour of hydrophobic peptides alone in aqueous solution due to the solubility issue, we integrated the potential MNP-helix as a part of rod-coil amphiphiles. To find the minimal length necessary for MNP-helix formation, we progressively increased the length of the peptides composed of alternating combinations of Ala and Aib ($α_m$), while the length of the hydrophilic coil segment (RGD) remained constant [$α_m$-(RGD)$_2$; Fig. 1c, d and Supplementary Figs. 1–3]. Initially, the RGD peptide was selected as a hydrophilic block because it is a well-known functional peptide targeting integrin receptors highly expressed in cancer cells[25]. Investigation by CD spectroscopy revealed a gradual increase in the α-helix content as the $m$ of the $α_m$ unit increased, and the formation of a nearly perfect α-helix was observed at $m = 10$ ($α_{10}$; Fig. 1d).

Because $α_m$-(RGD)$_2$ are amphiphiles with a self-assembling propensity, we could not confirm whether $α_{10}$ is a perfect α-helix in the monomeric state or whether the helix has been stabilized as a result of their assembly formation similar to that of coiled coils[26]. Indeed, atomic force microscopy (AFM) investigations showed that $α_m$-(RGD)$_2$ assembled into micelle-like spherical nanostructures (Supplementary Fig. 4). To exclude the possibility of assembly-induced helix stabilization, we designed PEG$_{16}$-$α_{10}$-PEG$_{16}$, which is unlikely to self-assemble due to the presence of bulky hydrophilic blocks at both ends of the hydrophobic $α_{10}$ unit. Here, PEG was used as a hydrophilic block for further in-depth physico-chemical studies using a more simplified molecule than the RGD peptide. Two distinct minima at 208 nm and 222 nm are the characteristic signatures of the α-helix (Fig. 1f). It should be noted that the PEG block does not have any CD signal (Supplementary Fig. 5). Moreover, the $[θ]_{222}/[θ]_{208}$ ratio was maintained at 1.1–1.2, indicating the nearly perfect α-helix formation (Fig. 1g). Typically, a $[θ]_{222}/[θ]_{208}$ ratio > 1.0 is found solely in interacting helices such in coiled coils[27]. However, the concentration independence of helix formation indicates that PEG$_{16}$-$α_{10}$-PEG$_{16}$ is monomeric (Fig. 1f, g)[23]. Moreover, AFM investigation visually verified that PEG$_{16}$-$α_{10}$-PEG$_{16}$ does not form discrete self-assembled structures, further supporting that the helices do not interact together (Supplementary Fig. 4d). To our knowledge, $α_{10}$ is the first MNP-helix reported to date.

### Magnetic control of self-assembly processes

Anisotropic diamagnetic molecules placed in isotropic solution can be aligned when a magnetic field is applied[12,28]. We investigated the magnetic alignment behaviour of the MNP-helix ($α_{10}$) by measuring residual dipolar couplings (RDCs) between $^1H$–$^{15}N$ amide pairs ($^1D_{NH}$) using nuclear magnetic resonance (NMR) spectroscopy at 16.45 T

(700.400 MHz) and 21.14 T (900.230 MHz). For NMR characterization, PEG$_{30}$-$α_{10}$-PEG$_{30}$ that has longer PEG chain compared to that of PEG$_{16}$-$α_{10}$-PEG$_{16}$, was designed to increase the solubility in organic solvent. It was confirmed that PEG$_{30}$-$α_{10}$-PEG$_{30}$ is also highly helical and monomeric (Supplementary Figs. 6–8). NMR spectra of PEG$_{30}$-$α_{10}$-PEG$_{30}$ were recorded in organic solvent (methanol-d3/chloroform-d) to make it sure that the molecule is in a monomeric state at the high concentration required for the NMR acquisition.

Magnetic anisotropy is defined as follows: $Δχ = χ_∥ − χ_⊥$, where $χ_∥$ is the magnetic susceptibility of the long axis and $χ_⊥$ is the magnetic susceptibility of the short axis. If $Δχ$ is positive, the molecule will align with the direction of the magnetic field and vice versa. It is known that α-helix orients parallel to the external magnetic field because $Δχ$ is positive[12]. We observed substantial N–H couplings ($^1J_{NH} + {}^1D_{NH}$) for all 21 amino acid residues of $α_{10}$ in PEG$_{30}$-$α_{10}$-PEG$_{30}$ (Fig. 2a, Supplementary Figs. 9–12, and Supplementary Tables 1 and 2). It is likely that resonances from two of the residues are close enough to appear as a single peak. The $Δχ$ correlates positively with the dipolar coupling ($^1D$)[29]. Thus, the positive signs of all experimental RDCs ($^1D_{NH, exp}$), ($^1J_{NH} + {}^1D_{NH}$)$^{(21.14\ T)}$ − ($^1J_{NH} + {}^1D_{NH}$)$^{(16.45\ T)}$ indicate that $α_{10}$ within the PEG$_{30}$-$α_{10}$-PEG$_{30}$ molecule aligns in the direction of the applied magnetic field (Supplementary Table 3)[29–31]. The magnitude of average experimental RDC value (2.037 Hz) of $α_{10}$ is fairly high as an organic molecule. In addition, the above results reveal that the peptide bonds in all 21 residues of $α_{10}$ are involved in the formation of the α-helix. Taken together, the results corroborate that the MNP-helix is magnetically responsive.

We then asked whether it is possible to magnetically control the self-assembly processes of the MNP-helix rod-coil building blocks. First, we investigated the ensemble-averaged solution behaviour of the self-assembled nanostructures under a magnetic field. To make it possible to observe this phenomenon, we designed a series of linear and cyclic rod-coils with different lengths of the α unit (Fig. 2b). Similar to $α_m$-(RGD)$_n$, CD analyses showed a gradual increase in the α-helix content as the $m$ of the $α_m$ unit increased from 3 to 8 in both L-$α_m$-PEG$_{10}$ and C-$α_m$-PEG$_{10}$ (Supplementary Table 4). To assess whether the helical stability of the $α_m$ unit in the molecular assembly is influenced by the magnetic field, L-$α_5$-PEG$_{10}$ and C-$α_5$-PEG$_{10}$, which have partially stable α-helices, were selected for further magnetic circular dichroism (MCD) investigation. Comparison of CD spectra in the absence or presence of a magnetic field revealed that the α-helix contents, i.e., $[θ]_{222}/[θ]_{208}$ ratios, were increased in both molecules when the magnetic field was applied to the samples in aqueous solution (Fig. 2c, d). Thus, the $α_m$ unit can make a conformational change to a more highly helical state when the magnetic field is applied.

Second, we tested whether it is possible to magnetically control the self-assembly process and the resulting self-assembled morphology. To induce the formation of self-assembled nanostructures, the rod-coils dissolved in aqueous solution were sonicated vigorously to disrupt the molecules and then incubated overnight for the molecules to be able to gradually assemble into stable nanostructures. To examine the effect of the magnetic field on self-assembly, we placed the sonicated molecule between the permanent magnets during the incubation period (Fig. 2e). We investigated the effect of magnetic field on the self-assembly in a series of cyclic and linear rod-coils (C-$α_m$-PEG$_{10}$ and L-$α_m$-PEG$_{10}$, $m = 3$–8). We progressively increased the length of the hydrophobic α-helical rod block while maintaining the length of hydrophilic PEG block. The experiments were performed, at which the concentrations of the building blocks were higher that their critical aggregation concentrations (CACs) (Supplementary Fig. 13 and Supplementary Table 5). When the $m$ was from 3 to 6, vesicle-like spherical assemblies[32–34] were formed both without and with the magnetic field (Fig. 2f and Supplementary Figs. 14 and 15). However, the magnetic field-induced morphological transformation of molecular assemblies could be observed when the $m$ was increased to 7 or 8. Especially, the

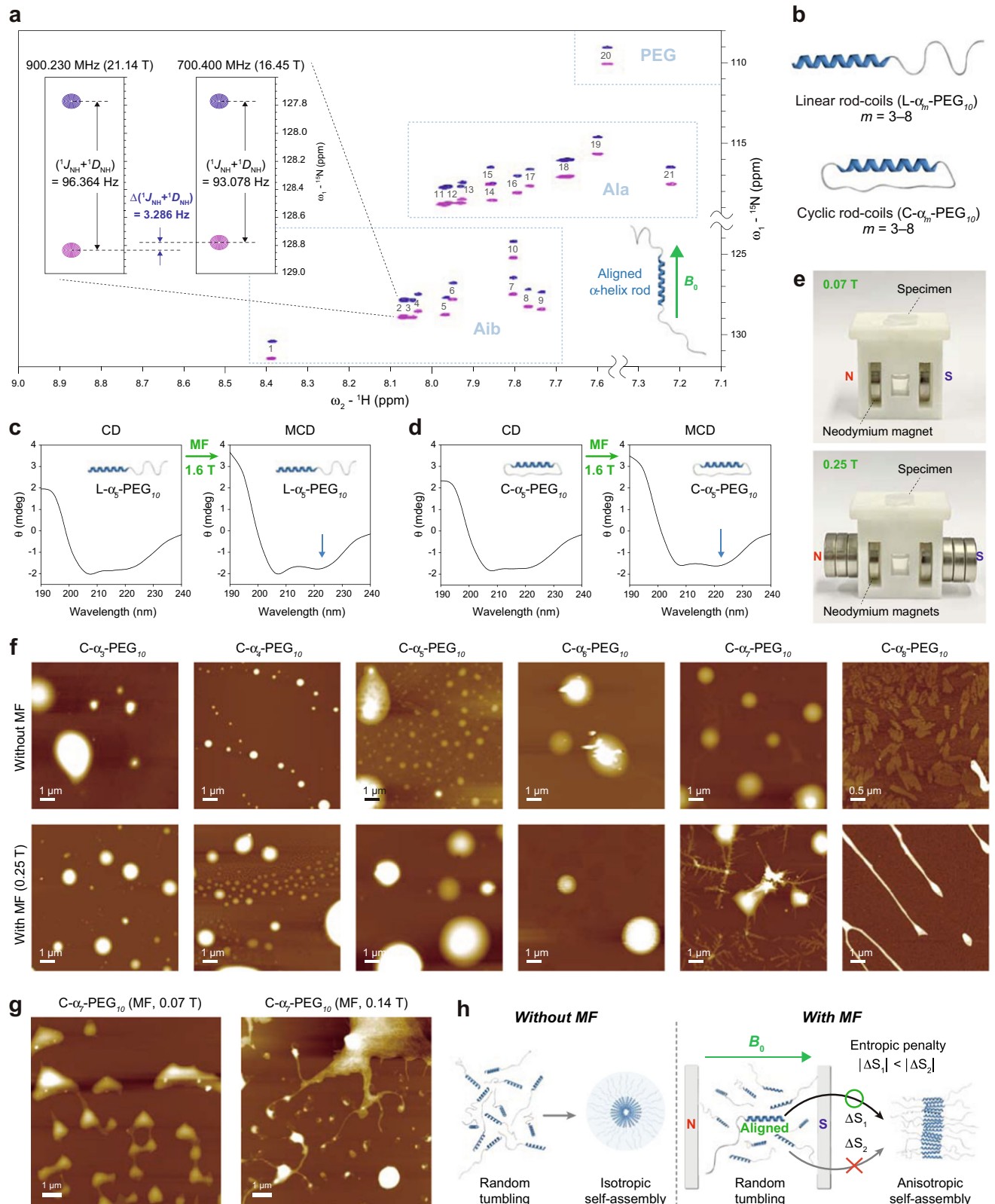

**Fig. 2 | Behaviours of the MNP-helix rod-coils under a magnetic field. a** The amide region of the 900 MHz $^1$H–$^{15}$N IPAP-HSQC NMR spectrum of PEG$_{30}$-α$_{10}$-PEG$_{30}$ in methanol-d3/chloroform-d (1:1) at 298 K, showing the alignment of the MNP-helix rod with the direction of the applied magnetic field. PEG: amide groups in PEG30 (see the Supplementary Fig. 1). **b** Design of self-assembling linear and cyclic rod-coils. **c** CD and magnetic circular dichroism (MCD) spectra of L-α$_5$-PEG$_{10}$ in water. **d** CD and MCD spectra of C-α$_5$-PEG$_{10}$ in water. **e** Placement of the

aqueous solution of the rod-coils (specimen) between neodymium magnets. Magnetic field strength was measured using a Model 450 Gaussmeter (Lake Shore Cryotronics, USA). **f** Effect of MF (0.25 T) on the self-assembly of C-α$_m$-PEG$_{10}$ ($m$ = 3–8). The peptide concentration was 20 μM, which is higher than the CAC. **g** Effect of MF strength on self-assembly of C-α$_7$-PEG$_{10}$. [C-α$_7$-PEG$_{10}$] = 20 μM. **h** Model of self-assembly processes in the absence or presence of the magnetic field.

transformation was most pronounced when the $m$ was 7 including the formation of cruciform anisotropic nanostructures, which are unlikely to be found in typical molecular assemblies (Fig. 2f and Supplementary Fig. 16). The level of transformation tends to be more highly affected as the strength of the magnetic field increased from 0.07 to 0.25 T (Fig. 2f, g). Magnetically induced transformation could not be observed at a concentration below CAC (Supplementary Fig. 17). The data indicate that a certain minimal length of the α-helical rod is required for magnetically responsive self-assembly.

Helix length dependence of morphological transformation was also observed in the linear rod-coils, L-$\alpha_m$-PEG$_{10}$ (Supplementary Fig. 18). The minimal length of the α-helical rod ($m = 4$) required for the transformation was shorter than that of the cyclic rod-coils. When the $m$ was more than 6, the linear rod-coils formed irregular aggregates rather than forming discrete assemblies, possibly due to the strong propensity to make hydrophobic interactions. Another type of linear rod-coil, $\alpha_{10}$-(RGD)$_2$, was also tested for magnetic transformation; however, we could not observe magnetic transformation with this molecule. In comparison, the magnetic transformation into anisotropic and unique planar nanostructures even at the weak magnetic field strength (0.07 T) was evident when the length of a hydrophilic block was increased in $\alpha_{10}$-(RGD)$_3$ (Supplementary Fig. 19). Together, the hydrophobic-hydrophilic balance, the length of the α-helical rod, and topology (cyclic or linear) is important for magnetically responsive self-assembly.

Restriction in the motional degree of freedom of the rod-coil molecules during the self-assembly process due to the alignment of the MNP helices along the magnetic field is likely responsible for the self-assembled morphology transformation. Considering that molecules with the $^1$H-$^{15}$N RDC values of ~20 kHz are considered to be fully oriented[35], only a small fraction of rod-coils should be aligned in solution considering the average experimental RDC value of $\alpha_{10}$ (Supplementary Table 3). However, the entropic penalty associated with self-assembly should be smaller for the aligned rod-coils than for the rod-coils randomly tumbling in solution (Fig. 2h and Supplementary Fig. 16b). Thus, the aligned rod-coils will participate in the self-assembly process more readily than the others, which would alter the overall dynamics of supramolecular crystal growth. This behaviour would have not been possible without the rod-like characteristics of the MNP-helix in a monomeric state.

## Cooperative, hierarchical, and living assemblies from the rod-coil peptide and DNA

Next, we aimed to develop magnetically responsive assemblies that have specialized functions. Duplex DNA has a large magnetic anisotropy value as an organic molecule due to the presence of base stacking[28,30,31]. DNA orients perpendicular to the external magnetic field because $\Delta\chi$ is negative for DNA[29]. Because the signs of $\Delta\chi$ for α-helix and DNA are opposite each other, it should be interesting to fabricate the supramolecular complexes of the MNP-helix rod-coil and DNA. Given the usefulness of rod-coils in the formation of well-defined assemblies with DNA such as filamentous virus-like particles[36,37], we designed modified MNP-helix rod-coils (R$_x$-$\alpha_{10}$-PEG$_{16}$, $x = 1$–4), that have positively charged arginine (Arg or R) residues at the distal end of the hydrophobic $\alpha_{10}$ rod (Fig. 3a). The $\alpha_{10}$-PEG$_{16}$ was designed as a basis of the above building blocks considering the hydrophobic-hydrophilic balance. Because α-helix and DNA respond to the magnetic field in different directions, we hypothesized that the overall magnetic responsiveness would be increased in the supramolecular complexes of R$_x$-$\alpha_{10}$-PEG$_{16}$ and DNA.

To visualize the progress of DNA condensation with ease, very large linear DNA consisting of 7528 bp was prepared by the single restriction digest of a plasmid DNA (Supplementary Fig. 20). Visualization with AFM confirms that the linear plasmid DNA is a hair-shaped

flexible molecule (Fig. 3b). The molecular assemblies of R$_1$-$\alpha_{10}$-PEG$_{16}$ were micelle-like but slightly irregular, possibly due to electrostatic repulsions in the micelle core (Fig. 3c and Supplementary Fig. 21). We then added increasing concentrations of R$_x$-$\alpha_{10}$-PEG$_{16}$ to a fixed amount of the linear plasmid DNA, and the formation of the peptide/DNA complex (PD-complex) was monitored using an electrophoretic mobility shift assay (EMSA). The gradual band retardation in EMSA as the charge ratio (+/−) increased is an indication of PD-complex formation (Fig. 3d and Supplementary Fig. 22). The band smearing is an indication of cooperatively assembled complex formation because multimers containing different number of building blocks can exist simultaneously in cooperative assemblies[36,38,39]. The building blocks containing higher number of positive charges could make DNA complexes more efficiently at low +/− ratios. The building block without positive charge, $\alpha_{10}$-PEG$_{16}$, was not able to make PD-complexes (Supplementary Fig. 23). These are evidences that electrostatic interactions between the peptide and DNA are crucial for the complex formation. R$_1$-$\alpha_{10}$-PEG$_{16}$ was selected for further in-depth studies among the four building blocks anticipating that weakly associated PD-complexes would be more magnetically responsive.

Microscopy studies revealed that the hierarchically ordered PD complexes of R$_1$-$\alpha_{10}$-PEG$_{16}$ and the linear plasmid DNA were formed in a slow and stepwise manner (Fig. 3e–h). AFM investigation revealed that PD complexes can be formed at charge ratios of as low as 0.7. We observed intermediate complexes in which R$_1$-$\alpha_{10}$-PEG$_{16}$ wrapped around the linear plasmid DNA (Fig. 3e). Approximately one day after the mixing of the peptide and DNA, rigid ribbon-like structures (nanoribbons) began to appear (Fig. 3f). Cooperative assembly is characterized by the nucleation and growth mechanism, in which the sequential assembly allows the formation of ordered structures. We frequently observed flat layers (terraces) and kinks in the nanoribbons, indicating that they are formed by the epitaxial growth mechanism of crystal formation. The presence of steps and kinks between terraces suggests that the nanoribbon growth can be described by the terrace-step-kink (TSK) model[40] of crystal surface formation. The height of the step coincided with the diameter of DNA (2 nm) and its multiples. Thus, the nanoribbon consists of multiple layers of flat PD complexes in a highly ordered state similar to solid-state crystals. The nanoribbons formed a stable colloidal suspension in aqueous solution.

The nanoribbon has a living character. In polymer chemistry, the term "living" refers to the ability of an already formed polymer chain to further react with a freshly supplied monomer. Living behaviour has been widely reported during the polymer synthesis, such as in living polymerization[41] and living supramolecular polymerization[42,43]. Recently, living behaviour has also been found to occur in self-assembly, such as in living crystallization-driven self-assembly (CDSA)[44–46] and the fibrillization of amyloid-β peptides[47,48]. Approximately 2 weeks after incubation, the living nanoribbons were further assembled into micrometre-scale objects, which we termed artificial chromosomes (Fig. 3g–i). The artificial chromosome was the final and global energy minimum state of the PD-complex because it was maintained after additional prolonged incubation. We also found that R$_4$-$\alpha_{10}$-PEG$_{16}$ could form the nanoribbons and artificial chromosomes with the linear plasmid DNA (Supplementary Fig. 24). CD analyses show that R$_1$-$\alpha_{10}$-PEG$_{16}$ and the linear plasmid DNA maintain the α-helical conformation and B-DNA structure, respectively, within the PD-complexes (Fig. 4a and Supplementary Fig. 25). The selected area electron diffraction (SAED) pattern and wide-angle X-ray scattering (WAXS) data further support that the α-helical conformation of the peptide and DNA double helix structure are maintained in both the nanoribbons and artificial chromosomes (Supplementary Figs. 26 and 27).

Overall, R$_1$-$\alpha_{10}$-PEG$_{16}$ and the linear plasmid DNA interact cooperatively to undergo a hierarchical assembly pathway with the

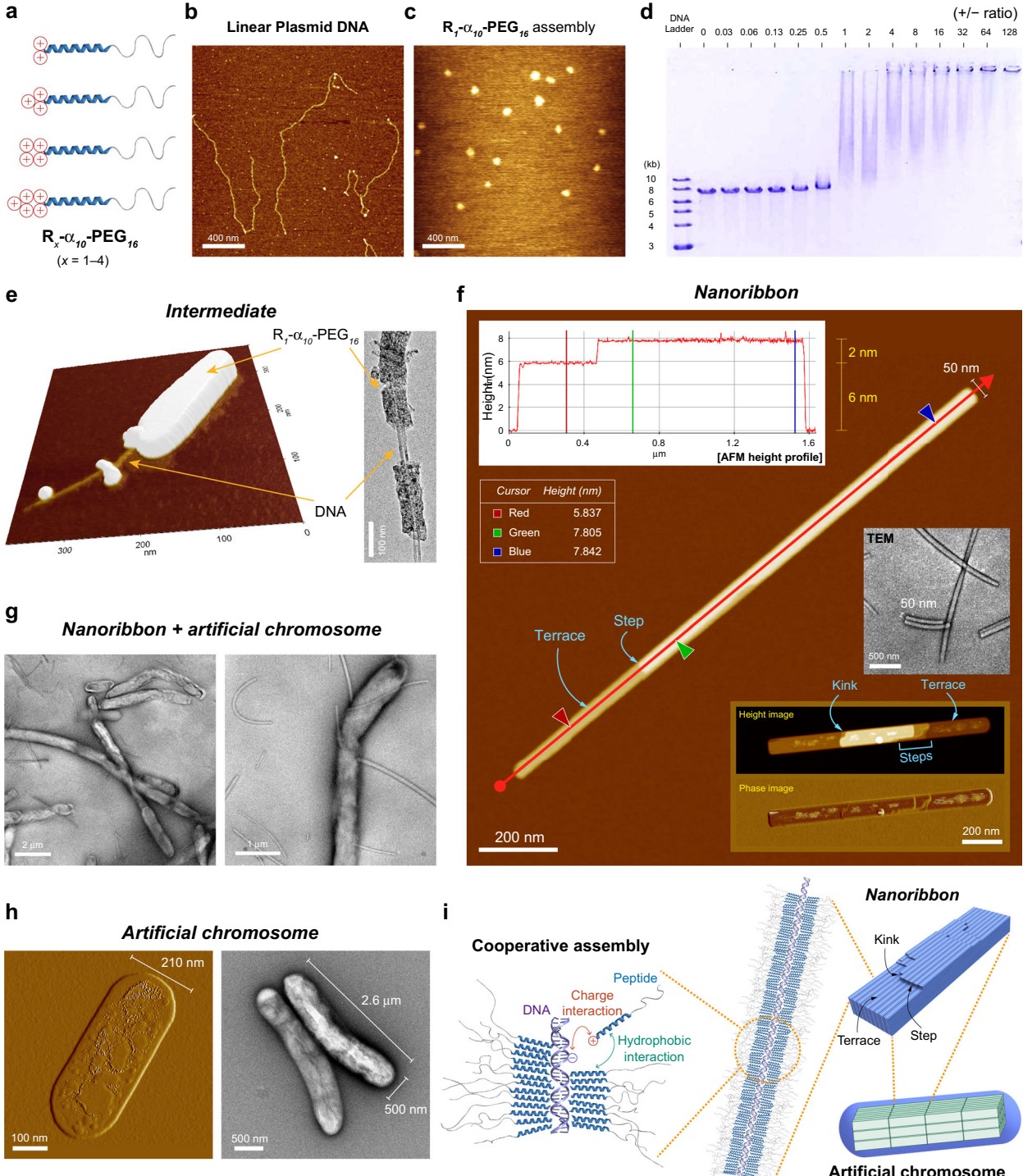

**Fig. 3 | Cooperative and living assembly of the rod-coil peptide and DNA.**
**a** MNP-helix rod-coils containing positively charged amino acid residues, $R_x$-$\alpha_{10}$-$PEG_{16}$ ($x = 1$–4). The building blocks have two to five positive charges. **b** A linear plasmid DNA (7.5 kb). **c** Self-assembled nanostructures of $R_1$-$\alpha_{10}$-$PEG_{16}$. **d** EMSA on agarose gel. **e** The PD complexes (intermediates). Left: an AFM image, right: a TEM image. The charge ratio (+/−) was 0.7. **f** The PD complexes (nanoribbons). **g** The PD complexes (nanoribbons + artificial chromosome). **h** The PD complexes (artificial chromosomes). **i** The molecular mechanism responsible for the formation of the nanoribbon.

following notable transitions, cooperative assembly of the DNA and the rod-coil peptide → (living) nanoribbon → artificial chromosome (Fig. 3i). During the assembly, both charge interactions between the arginine residue and DNA and hydrophobic interactions between the MNP-helices act in a cooperative manner. The formation of assemblies with crystalline cores and solvated coronas should be responsible for the living characteristics of the nanoribbon[46].

**Magnetically responsive disassembly of artificial chromosomes**
The structure and the assembly mechanism indicate that DNA is tightly packaged into the artificial chromosome with the help of $R_1$-$\alpha_{10}$-$PEG_{16}$, which shares similarity with human chromosomes. Approximately 2 m of DNA inside the small cell nucleus (~6 μm in diameter) is packaged tightly into the chromosome and chromatin with the help of positively charged histone proteins, which otherwise would take up a very large

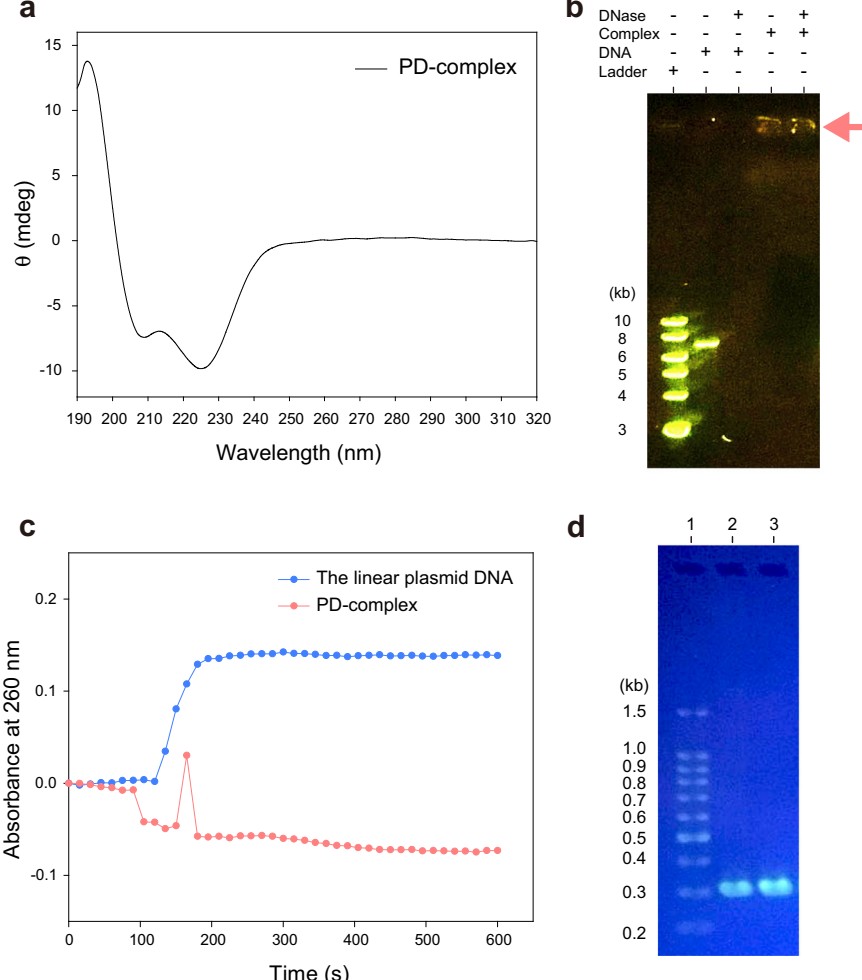

**Fig. 4 | Stability, DNA data storage, and information recovery, and magnetically responsive disassembly of the artificial chromosomes. a** CD difference spectrum of PD-complex. The PD-complex was fabricated using $R_I$-$\alpha_{10}$-$PEG_{16}$. **b** DNase protection assay. Ladder: DNA ladder, DNA: linear plasmid DNA, Complex: PD-complex, DNase: DNase I. An arrow indicates the PD-complexes trapped in the well (lanes 4 and 5). **c** Time-dependent DNase protection assay. **d** Recovery of information stored in DNA using polymerase chain reaction (PCR). Lane 1, DNA ladder; lane 2, PCR-amplified DNA from the linear plasmid DNA; lane 3, PCR-amplified DNA from the artificial chromosome.

space. The artificial chromosomes are also very similar to human chromosomes (length: ~2–13 μm) in dimension and shape. In the era of big data, the development of a method to store rapidly expanding data in minimal space while enabling convenient data recovery is urgent. The DNA-based data storage method has been considered a promising alternative to conventional methods[49–52]. The artificial chromosome can become a promising DNA data storage medium because it offers a very high DNA packaging density.

DNA data storage media also need to protect DNA from degradation, because DNA, in its naked state, is prone to enzymatic, chemical, and physical damage[53]. The DNase I protection assay showed that DNA could be protected from nuclease attack by the formation of the artificial chromosome (Fig. 4b). A time-dependent DNase protection assay quantitatively demonstrated that DNA, in its naked state, degraded rapidly upon the addition of DNase I; however, DNA, when it was incorporated into the artificial chromosome, was completely protected from nuclease digestion during the prolonged incubation period (Fig. 4c and Supplementary Fig. 28). We attempted to retrieve the DNA sequence information from the artificial chromosome. We found that the amplification of a specific fragment of the linear plasmid DNA could be achieved using the standard PCR protocol even when the DNA was incorporated into the artificial chromosome (Fig. 4d). Thus, $R_I$-$\alpha_{10}$-$PEG_{16}$ molecules and the

supramolecular assembly do not interfere with the DNA amplification processes.

Next, we asked whether the disassembly of the PD-complexes can be induced by the magnetic field. We found that the magnetically induced disassembly was indeed possible, but the time required for the disassembly was fairly long (~2 weeks) when the PD-complexes were exposed to a static magnetic field (0.1 T) (Supplementary Fig. 29). To determine if the time required for the disassembly can be decreased by using a rotating magnetic field (RMF), we exposed the nanoribbons (Fig. 5a) under RMF (0.25 T, 1200 rpm) for 1 h (Fig. 5b). TEM investigation of the nanoribbons after the exposure to RMF shows that most of the nanoribbons have been disassembled into the peptide aggregates, the linear plasmid DNA, and the nanoribbons in the middle of degradation (Fig. 5c–i). Some of the DNAs exist as multiple DNA strands (Fig. 5f, h) or the nanoribbons have been fully disassembled into a single DNA strand (Fig. 5g). It could be observed that several DNA strands lie parallel to each other (Fig. 5h, i), which, conversely, supports the cooperative assembly mechanism of the nanoribbon (Fig. 3f, i). The opposite signs of $\Delta\chi$ between α-helix and DNA would have facilitated the magnetically induced unpacking of DNA from the artificial chromosomes (Fig. 5i). The nanoassemblies of $R_I$-$\alpha_{10}$-$PEG_{16}$ are also affected by RMF (Supplementary Fig. 30). It should be noted that the peptide and DNA remained intact after the exposure to RMF

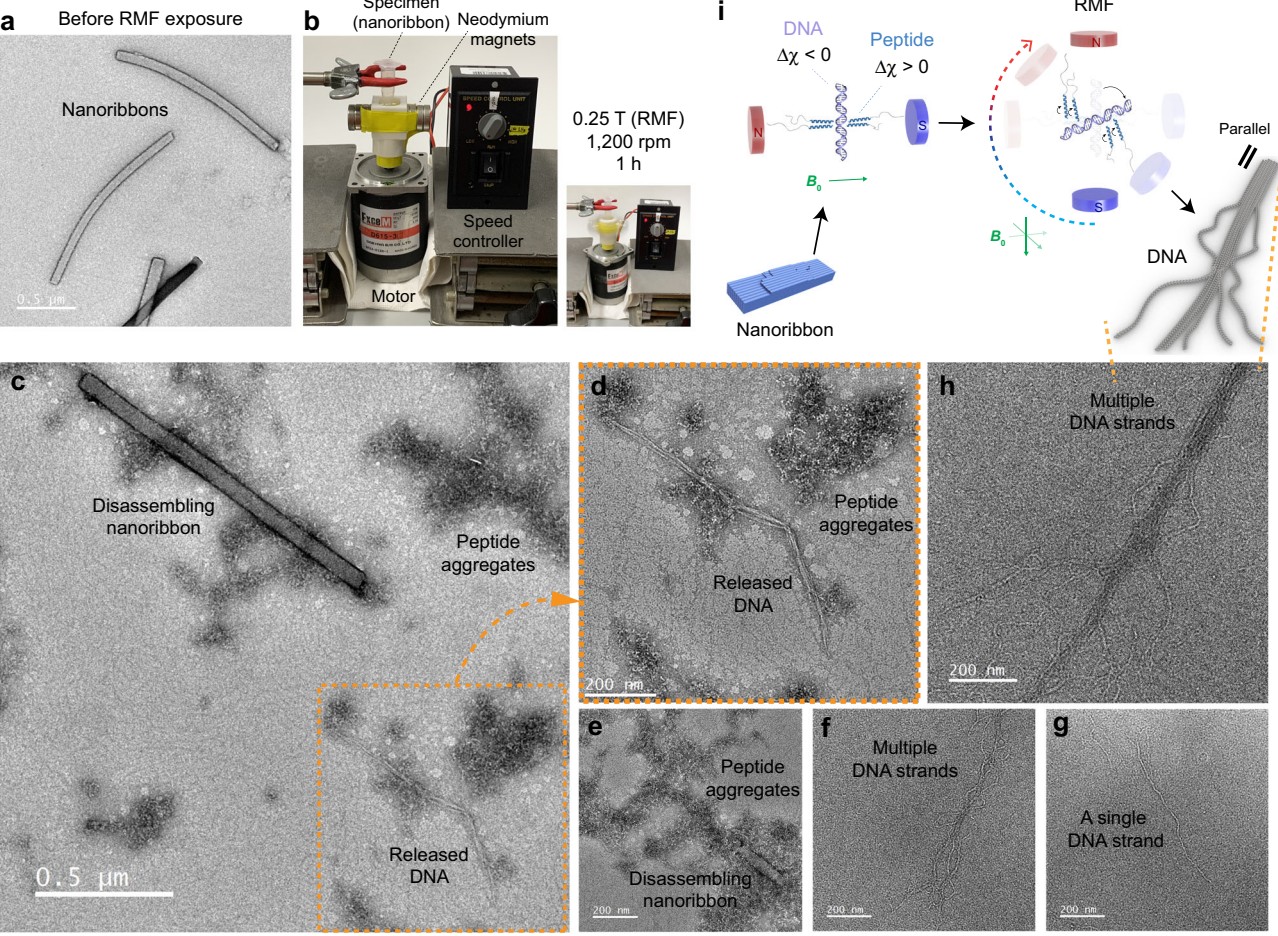

**Fig. 5 | Magnetically responsive disassembly of the artificial chromosomes.**
**a** TEM image of the nanoribbons before the exposure to magnetic field. RMF = rotating magnetic field. **b** An in-house equipment to generate an RMF from permanent magnets. **c**–**i** Disassembly of the artificial chromosome by the rotating magnetic field.

(Supplementary Fig. 31). Taken together, the time required for the magnetically induced disassembly can be controlled by changing the field strength and/or the type of magnetic field.

## Discussion

The study shows that the magnetic control of assembly and disassembly under the ordinary magnetic field strength is possible by synergistically combining the concepts of the perfect α-helix and the rod-coil supramolecular building blocks. We have experimentally verified the magnetic responsiveness both in solution state and in thin films containing equilibrium morphologies. To make this possible, the key prerequisite was the development of the α-helical peptide, which has multiple characteristics simultaneously. The MNP-helix, which is compact in size, highly helical even in a monomeric state, and hydrophobic, satisfied the required conditions. By using the MNP-helix as a rod block, we showed that the self-assembly processes of the designed rod-coils can be magnetically controlled to produce unique molecular assemblies that are usually not observable when fabricated with typical self-assembly processes.

Hierarchical and living self-assembly between the positively charged rod-coil and DNA exemplifies the formation of highly complex and exquisite molecular assemblies, which share a mechanistic similarity to living CDSA and the amyloid-β assembly. In chromosomes, elongated and fluttering DNA chains are hierarchically condensed into compact structures in which DNA is wrapped around histone proteins. DNA is also highly condensed into the compact structures in the artificial chromosome; however, the formation mechanism is different

from that of biological chromosomes because DNA strands lie parallel to each other in the artificial chromosomes. This study lays foundation to magnetically interface organic materials with magnetic devices and instruments, with application potentials in magnetically responsive bionanomaterials, molecular magnetic devices, and smart peptide/nucleic acid complexes highly responsive to magnetic field.

## Methods

### Peptide synthesis

The peptides were synthesized manually using the Fmoc solid-phase peptide syntheses (SPPS) protocol on Rink Amide MBHA resin LL (100–200 mesh, 0.30–0.40 mmol g$^{-1}$, Novabiochem, Germany). All Fmoc-amino acids, Fmoc-NH-PEG8-propionic acid, and Fmoc-NH-PEG10-propionic acid were purchased from AAPPTec (USA). Coupling reagents such as 2-(6-chloro-1H-benzo-triazole-1-yl)−1,1,3,3-tetramethylaminium hexafluorophosphate (HCTU) and 1-hydroxybenzotriazole (HOBt), and Fmoc-PEG2-Suc-OH (Fmoc-ebes-OH) were purchased from AnaSpec (USA). The synthesis scale was typically 0.1 mmol, and the synthesis was conducted in a 6 mL Resprep solid-phase extraction (SPE) tube (Restek, USA). The peptides were purified by high-performance liquid chromatography (HPLC) on a C4 reversed-phase column (Waters, USA) at room temperature. The eluents were water (0.1% TFA) and acetonitrile (0.1% TFA). The molecular weight of the peptide was confirmed by matrix-assisted laser desorption/ionization time-of-flight (MALDI-TOF) mass spectrometry. The purity of the peptide was >95%, as determined by analytical HPLC.

## Circular dichroism (CD) spectroscopy

To evaluate the structures of the peptides and the linear plasmid DNA, CD spectra were obtained using a Chirascan CD spectrometer (Applied Photophysics, UK). The concentration of the peptides typically ranged from 0.156 to 160 μM. The final concentration of the linear plasmid DNA was 30 ng/μL. All the samples were dissolved in $H_2O$. For the formation of the artificial chromosome, the peptide and the DNA mixture was incubated for more than 12 days. Measurements were performed in a 2-mm path-length cuvette. Each scan was repeated three times and averaged. Mean residue ellipticity (MRE) was calculated based on the number of amino acid residues.

Magnetic circular dichroism (MCD) spectra were obtained using a Jasco-815 159-L spectrophotometer (Jasco, Japan). The concentration of the peptides was 20 μM in water. Measurements were performed in a 2-mm path-length cuvette. The magnetic field of the permanent magnet in the MCD accessory (PM-491) was 1.6 T. Each scan was repeated three times and averaged.

## Nuclear magnetic resonance (NMR) spectroscopy

[1]H NMR and 2D [1]H-[15]N IPAP-HSQC (in-phase/anti-phase heteronuclear single quantum coherence) NMR spectra were acquired at 298 K using a Bruker AVANCE Neo 900 MHz spectrometer and a Bruker AVANCE III HD 700 MHz spectrometer, both equipped with cryogenic probes. The NMR samples of $PEG_{30}$-$α_{10}$-$PEG_{30}$ were dissolved in methanol-d3/chloroform-d (1:1) containing 0.05% TFA at a concentration of 4.26 mM. The acquisition parameters of IPAP were 2048 $t_2$ × 1024 $t_1$ points, 8 scans, 0.3 s recycle delay, 14 ppm spectral width for [1]H and 26 ppm spectral width for [15]N. The acquired data were split into two data sets, IP and AP, using a TopSpin software (Bruker, Germany). Each spectrum was processed by zero-filling up to 8192 × 8192 points. Peak positions were assigned by iterative integration using an NMRFAM-SPARKY software.

## Atomic force microscopy (AFM)

AFM was performed using the NX10 system (Park Systems, Korea) in non-contact mode with PPP-NCHR AFM probes (Nanosensors, Switzerland). The DNA concentration ranged from 1 to 30 ng/μL, and the peptide concentration ranged from 1 to 32 μM in $H_2O$. Two microliters of the sample solution were cast onto a freshly cleaved mica surface and dried. The data obtained by the SmartScan program (Park Systems, Korea) were processed and analysed using the XEI program (Park Systems, Korea).

## Transmission electron microscopy (TEM) and electron diffraction

TEM was performed using a JEM-F200 multi-purpose electron microscope (JEOL, Japan) at 200 kV. Two microliters of the sample solution was loaded on a copper grid (carbon type-B grid or Formvar/silicon monoxide grid, 200 mesh copper grids with a 97 μm hole; Ted Pella, USA). One hour after loading, 2 μL of 1–2% uranyl acetate was added to the dried sample for less than a minute, and the excess negative stain solution was wicked off by filter paper. To analyse the internal packing structures of the artificial chromosomes, electron diffraction (ED) patterns of selected areas were acquired.

## Linear DNA preparation

We used the linearized plasmid DNA of sufficient length (estimated length: ~2.6 μm) for facile structural observation with AFM and TEM. Large-scale plasmid DNA (pTWIN-Rev; 7,528 bp) preparation was performed using the NucleoBond Xtra Maxi Plus kit (Macherey-Nagel, Germany). The plasmid DNA was digested into linear double-stranded DNA by BamH1 restriction enzyme (New England Biolabs, USA). The linear plasmid DNA was purified from the restriction digest with 1.1× agarose gel. Electrophoresis was conducted at 120 V for 180 min in 1× TBE buffer. After electrophoresis, the DNA was stained with SYBR Safe DNA Gel Stain (Invitrogen, USA) for visualization. A slice of the DNA band that corresponds to 7.5 kb linear dsDNA was cut out, and the linear plasmid DNA was extracted using a MEGA quick-spin™ plus kit (iNtRON, Korea). The purity of the DNA was found to be high ($A_{260}$/$A_{280}$ ratio was 1.8 and $A_{260}$/$A_{230}$ ratio was 2.0–2.2) as measured by a Nanodrop 1000 spectrophotometer (Thermo Scientific, USA).

## Formation of the artificial chromosomes

Artificial chromosomes were formed by slowly injecting the peptide $R_1$-$α_{10}$-$PEG_{16}$ (34 μM in $H_2O$) into linear plasmid DNA (30 ng/μL in $H_2O$) in equal volumes. Before mixing commenced, the peptide solution was sonicated extensively to disrupt potential nonspecific aggregates. The solution was further mixed by gentle pipetting, vortexed for an hour, and incubated at room temperature for a prolonged period. Occasionally, 2 μL of the mixture was sampled, and the morphological states were characterized by AFM and TEM.

## Electrophoretic mobility shift assay (EMSA)

The gel electrophoretic mobility shift assay was performed to investigate the mode of interactions between R-$α_{10}$-$PEG_{16}$ and the linear plasmid DNA and their binding ratios. The concentration of the peptide was increased from 1.5 to 6214 μM, while the concentration of the linear plasmid DNA was fixed at 30 ng/μL. The same mixing protocol was used as in the artificial chromosome formation. After the incubation at room temperature, 10% glycerol was added to the sample for gel loading. Electrophoresis was performed for 100 min at 90 V in a 1% agarose gel. The bands were visualized by staining with SYBR Safe DNA Gel Stain (Invitrogen, USA).

## DNase I protection assay

A sample (the linear plasmid DNA or the artificial chromosome) containing 60 ng (2 μL; 30 ng/μL) of DNA was mixed with 0.1 μL (0.2 units) of DNase I and 1 μL of 10× DNase I reaction buffer. The volume of the digestion mixture was 10 μL. After 20 min of incubation at 37 °C, 0.3 μL of 0.05 M EDTA was added to inactivate the enzyme. Then, the sample was mixed with 1.7 μL of 60% glycerol and electrophoresed for 120 min at 100 V on a 1.1% agarose gel. For visualization, DNA was stained with SYBR Safe DNA Gel Stain (Invitrogen, USA). To evaluate the time-dependent kinetics of DNA degradation, UV absorbance at 260 nm was recorded over time after the addition of DNase I. Each sample (71.2 μL) containing 30 ng/μL of DNA and 10× DNase I reaction buffer (8 μL) was mixed in a cuvette. Then, DNase I (0.8 μL) was added to the cuvette, and the sample was mixed thoroughly by pipetting for ~90–105 s. The measurement of absorbance ($A_{260}$) was performed in a V-650 UV–vis spectrophotometer (JASCO, Japan) using a 10-mm path-length quartz cuvette. The absorbance was recorded at intervals of 15 s.

## Information recovery from the artificial chromosomes

To assess the capability of recovering the DNA information stored in an artificial chromosome, a polymerase chain reaction (PCR) was conducted to amplify a specific DNA segment of the linear plasmid DNA. The specific 310 bp segment was amplified using the sense primer 5′-ACG GAG ACT GGA GTC GAA GAG G-3′ and the antisense primer 5′-GTA GGG CAA CTA GTG CAT CTC CC-3′. Each reaction mixture (50 μL) contained 50 ng of DNA, 10 pmol of each primer, and 25 μL of 2× Quick Taq HS DyeMix (Toyobo, Japan). PCR amplification (30 cycles) was performed with a T100 Thermal Cycler (Bio-Rad, USA). Each cycle included an initial denaturation step at 94 °C for 2 min, a denaturation step at 94 °C for 30 s, a primer annealing step at 58 °C for 30 s, and a extension step at 68 °C for 1 min. The $T_m$ of the sense primer was 65.9 °C and the $T_m$ of the antisense primer was 66.4 °C. After the 30 cycles of amplification, the final extension step was conducted at 68 °C for 5 min. Then the samples were stored at 4 °C until use. The amplified products were investigated by agarose gel electrophoresis. The electrophoresis was conducted for 120 min at 80 V in a 2 % agarose gel

using 1× TBE buffer. SYBR Safe DNA Gel Stain (Invitrogen, USA) was used for staining.

## Data availability

All data supporting the findings described in this manuscript are available in the article and Supplementary Information. All other data are available from the corresponding author on request. Additional data on the disassembly of the PD-complexes have been deposited in figshare (https://doi.org/10.6084/m9.figshare.22829153).

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

## Acknowledgements

This work was supported by grants from the National Research Foundation (NRF) of Korea (2022M3E5F1016877 and 2022M3H4A1A0204645 to Y.L.). We thank Korea Basic Science Institute (KBSI) and Pohang Accelerator Laboratory (PAL) for NMR and X-ray diffraction (XRD) experiments, respectively.

## Author contributions

Y.J. and H.K. designed, performed, and analysed experiments and wrote the manuscript. H.C. performed NMR experiments and analysed the NMR data. Y.L. designed the project, supervised the research, and wrote the manuscript.

## Competing interests

Y.L., Y.J., and H.K. have filed a patent for the development of the MNP-helix and magnetically responsive supramolecular building blocks. Y.L. is a co-founder and equity holder of ArtProBio, Inc. The remaining author declares no competing interests.
