## [Peer Review File · Nature Communications]

Magnetic control of self-assembly and disassembly in organic materialsReviewers' Comments:

Reviewer #1:

Remarks to the Author:

The manuscript by Jung et al. reported the construction of a Rod-Coil peptide nanostructure that responsive to magnetic field. They showed that by carefully designing peptide building blocks, the self-assembly of rod-coil amphiphiles can be controlled in a magnetically responsive manner. They further fabricated artificial chromosomes by coassembly of rod-coil peptides and DNA. Upon magnetic stimulation, the artificial chromosomes disassemble into peptides and DNA. Overall, I think the construction of magnetically responsive assemblies is appealing, however, the data quality is poor and the interpretation of data was insufficient. E.g. important controls were missing. Thus, I cannot recommend the acceptance of this work. Detail comments:

1. It is quite obvious that α 10-RGD3 can form vesicles without MF; how would the authors explain C- α 5-PEG10 formed vesicles (Fig.2g)? In addition, AFM imaging in air cannot prove that the spherical nanoparticles were vesicles, as the salts and solute in the solution during the drying process will also result in nanoparticles. I suggest the authors measure the vesicles with DLS in the solution phase or conduct AFM imaging in liquid.
2. Without MF, C- α 5-PEG10 formed large rods, the authors should illustrate and explain the underlying mechanism.
3. In Fig.2c, the authors designed and studied the properties of linear and circular amphiphiles, however, when studying assembly behaviors, they chose C- α 5-PEG10 and α 10-RGD3, why? The α helix lengths and the hydrophilic segments were different (5 vs 10, and PEG vs RGD3), I feel this part is very weak. The authors should provide more data to explain how α helix length and hydrophilic segments affect the assembly process.
4. The authors claimed that they fabricated the artificial chromosomes. What are the driven forces of the assembly of artificial chromosomes? Is it the electrostatic interactions or the hydrophobic interactions between α helices? Only two positive charges and the weak hydrophobic interactions seemed difficult to package DNA as the living system did.
5. The authors used R- α 10-PEG16 to compact DNA, why chose α 10 and PEG16? The assembly and disassembly of solely R- α 10-PEG16 should be studied as it is very important what kind of supramolecular structures it will form.
6. Important controls, such as α 10-PEG16 without positive charge also should be provided in Fig. 5
7. Supplementary Fig. 2~3 should be mentioned in the main text.

Reviewer #2:

Remarks to the Author:

The authors report on exploiting the positive magnetic susceptibility of alpha helices to magnetically control assembly/disassembly of rod-coil molecules. They developed a library of rod-coil and coil-rod-coil molecules where the coil segment is either RGD or a PEG unit. CD was used to determine the helicity while AFM was used to determine the morphology of higher order structures. Finally, the authors designed one rod-coil monomer bearing a single Arginine to package DNA and demonstrate that under exposure to a rotating magnetic field they can disassemble these structures.

While I find the concept of utilizing the magnetic susceptibility of peptide motifs and self-assembly to potentially create functional structures interesting, I do not think this particular work demonstrates it well. The design space is limited, there is a lack of sufficient hypotheses, missing experimental evidence for the claims made, and the application is not exciting. There are lots of gaps to fill before this can be considered for publication in a high level and broad audience journal such as Nature Communications. To help the authors understand what is missing, I outlined detailed comments below:

- (1) Molecular design of rod-coil monomers:

The biggest criticism I have relates to the rationale behind the monomer library and how these choices were made. In the current form of the manuscript, most choices seem random, scattered, with no hypothesis or sufficient data to support it. It also does not inform the scientific community on what is needed or is critical/important to design similar materials or enhance their magnetic susceptibility.

Specific comments below:

- (1.1) Most design choices are not explained or supported by sufficient data. The authors used alternating A-Aib - would another order work too? What design parameters are most important?
- (1.2) Why RGD the coil domain and how was its length determined? - this choice seems random and is not explained or rationalized.
- (1.3) AFM is not sufficient to determine that monomers are not assembling. DLS is required to see that these are indeed monomers in the experimental conditions and don't form higher-order structures.
- (1.4) If the magnetic field can induce the peptide to form a stable alpha helix, it is not clear why starting with alpha10 versus alpha5 is better for all the other work presented in the paper.
- (1.5) From RGD on one end - coil was changed to a peg16 unit on both ends. Authors claim it is "to avoid self-assembly". Again - why peg16 versus any other peg length. Why capping the rod at both sides? Would one be sufficient and which one? Later monomers in the paper have only one peg arm - so this point seems important to demonstrate.
- (1.6) The authors claim the peg units have no role in stabilizing the helix formation of the peptide, but this claim is not supported by data. In fact, peg molecules are known themselves to form helices and coils (Macromolecules 2005, 38, 9333-9340), so it's not clear how this effect can be decoupled.
- (1.7) Next, for NMR it was changed to Peg30... but no CD or AFM characterization was done for this one or compared to the peg16 one or explanation provided why peg16 could not be explored with NMR!
- (1.8) Then, for the self-assembly - they explored linear and cyclic rod-coils - but again without any rational or hypothesis driving this design choice. And once again - peg unit length was changed to peg10 - no explanation!!
- (1.9) Self-assembly changes were only explored for Cyclic-alpha5-peg10 and linear-alpha10-(RGD)3 - why these ones, what happens for the others, and why? Can this behavior be predicted or generalized?
- Linear-alpha10-(RGD)3 seems particularly random - why coming back to RGD again? and this time with 3 RGD repeats? Why does this one behave differently than any other alpha-RGD monomer in the library?
- (1.10) Why C-alpha5-peg10 needs 0.25T and linear-alpha10-(RGD)3 requires 0.07T for the self-assembly switch?
- (1.11) For the application - the authors chose Arg-alpha10-PEG16 - why this one versus all the other monomers they tested? Why only one Arginine? Comparison between linear and cyclic monomers will be interesting to see if package the DNA differently.

(2) Assembly/Disassembly

The authors seem to "fish" for finding an interesting effect instead of being able to design for one. There is also insufficient evidence how (it at all) it is the magnetic field that is responsible for the effect seen. Specific comments below:

- (2.1) There are multiple assembly goals that should be investigated separately and explored via multiple techniques. One is unfolded helix to helix using magnetic field - the authors briefly show that for cycle alpha 5 - the magnetic field seems to be inducing the incomplete helix into a closely perfect one - this is an interesting effect that should be explored further. Second is the monomer to assembled state - I did not see results demonstrating the magnetic field can induce self-assembly of non-assembled monomers - this will be of high value to achieve. Third is the change of assembly from one morphology to the other using a magnetic field - the authors showed some evidence that this might be possible - but only for 2 of their monomers, and their sequences seem random. Authors should expand their investigation into additional monomers and explore how each domain (rod versus coil), (linear versus cyclic), experimental conditions, etc. affect this behavior. Also - authors will benefit from writing their manuscript with this above rationale of how I explained the various assembly behaviors explored in this manuscript for better clarity.
- (2.2) The magnetic field effect on self-assembly should be explored across a range of concentrations of the monomers. CMC/CAC should be established for each monomer to determine the concentration of assembly and magnetic effect on assembly should be explored below and above the cmc.

(3) Application

Current application is not particularly exciting and its performance is not convincing. Specific comment below:

(3.1) Again - choice behind why using R-Alpha10-Peg16 is not clear - see point 1 above. (3.2) Stability against degradation was only monitored for 600 seconds - this is not exciting for a real world application and should be evaluated for at least several hours to be meaningful. (3.3) Based on point 3.2 - 2 weeks were required for disassembly - how can we know if degradation of either DNA or peptide are not the driver for this?

(3.4) Even with the rotating magnetic disassembly - no evidence that DNA and peptides remained intact or that they can be repackaged in the absence of a magnetic field. (3.5) No evidence that DNA is indeed separate from peptides and can be fully functional. (3.6) Packaging of DNA under a magnetic field should be done to see if it affects the process.

Reviewer #3:

Remarks to the Author:

The manuscript by Lim and coworkers describes the magnetic-field induced assembly of especially designed helical amphipathic peptides, also in combination with DNA, where they result in assemblies as "artificial chromosomes", which can also be disassembled in turn, most efficiently by rotating magnetic fields. The conclusions in the manuscript are supported mainly by AFM and TEM, but also NMR spectroscopy on the monomeric peptides as proof for their helical nature. With respect to the latter, a chemical shift analysis could support the alpha-helical nature of the monomeric peptide in addition to RDCs. Also, only 21 resonances are observed in the NMR spectrum; however, the chemical structure shows several more NH pairs. Why are they not observed, and what are the assignments of the 21 peaks? Or at least peak groups?

Other remarks:

p19 line 8: there does not seem to be DNA condensation in this experiment (as with histones or protamines), rather only DNA wrapping? So the argument that this system shows very high DNA packaging density does not hold? Or please make a better argument to show that the DNA is condensed (volume in isolation, volume with amphipathic peptide assemblies?).

p19 line 20: it is surprising that a standard PCR protocol allows to amplify this protected plasmid - which chemical added, or other step in the protocol, could allow this? Or was the DNA disincorporated before?

p20 line 9: why is it expected that the assemblies disassemble at all in a magnetic field, which initially is used to induce them?

In general, the conclusions are difficult to understand for a non-specialist reader. For instance "Given the usefulness of rod-coils in the formation of well-defined assemblies with..." - useful for what? Or: "The band smearing that occurred at a charge ratio of 1 was an indication of cooperatively assembled complex formation^{32,34,35}." - here should be shortly explained how these references support this conclusion. There are other specialist shortcuts in the manuscript, and an effort to better discuss the findings in the light of the references given would be in general useful for the reader.

Reviewer #4:

Remarks to the Author:

The work of Lim et al., reports on the synthesis of alpha-helix-peptides-PEG block-copolymers with

enhanced respond to magnetic fields. This offers the possibility to tune self-assembly processes remotely by the application of an external magnetic field. The system developed by these authors has the ability to respond to mild external magnetic fields which make applications more accessible. Additionally authors tested a selected alfa-peptide-PEG copolymer by its capacity to interact electrostatically with a DNA strand to form a supramolecular structure that is able to protect the DNA for enzymatic degradation and simultaneously do not avoid its replication. Moreover the peptide-PEG covering can be removed by the application of an external magnetic field. This work is very innovative, has been well-conducted, the data is clearly presented and results are very convinced. This work has important implications in fields of magnetic response materials, supramolecular systems, smart materials and biomaterials, among others. I recommend its publication as it is.

Reviewers' comments and our responses

Reviewer #1:

(Comment 1)

The manuscript by Jung et al. reported the construction of a Rod-Coil peptide nanostructure that responsive to magnetic field. They showed that by carefully designing peptide building blocks, the self-assembly of rod-coil amphiphiles can be controlled in a magnetically responsive manner. They further fabricated artificial chromosomes by coassembly of rod-coil peptides and DNA. Upon magnetic stimulation, the artificial chromosomes disassemble into peptides and DNA. Overall, I think the construction of magnetically responsive assemblies is appealing, however, the data quality is poor and the interpretation of data was insufficient. E.g. important controls were missing. Thus, I cannot recommend the acceptance of this work.

(Response 1)

We would like to thank the reviewer for thoughtful and helpful comments on our manuscript. We would like to make responses to the reviewer's comments. Revised phrases and sentences are highlighted in blue in the manuscript and the supplementary information.

We would like to point out that, before deciding to show some of the results from selected building blocks, we already have synthesized many more building blocks and have performed a lot of control experiments with those building blocks that was not illustrated in the original manuscript. During the preparation of the original manuscript, we did not fully explain the results from those control building blocks to avoid an increase in manuscript length. Having commented about the lack of sufficient controls from the reviewer, we now provide more data in the revised manuscript to explain why we selected some of the building blocks for the magnetic responsiveness and DNA packaging. For more details, please see the Response 4.

(Comment 2)

It is quite obvious that α 10-RGD3 can form vesicles without MF; how would the authors explain C- α 5-PEG10 formed vesicles (Fig.2g)? In addition, AFM imaging in air cannot prove that the spherical nanoparticles were vesicles, as the salts and solute in the solution during the drying process will also result in nanoparticles. I suggest the authors measure the vesicles with DLS in the solution phase or conduct AFM imaging in liquid.

(Response 2)

We would like to make a correction that C- α 5-PEG₁₀ mentioned in the original manuscript was C- α 7-PEG₁₀. We correct this error in the revised manuscript. Appropriate data related to characterization of C- α 5-PEG₁₀ are supplemented in the revised manuscript (Fig. 2f).

Considering the reviewer's comment about how C- α 7-PEG₁₀ would form vesicles, we added a plausible model regarding the vesicular self-assembly of C- α 7-PEG₁₀ as Supplementary Fig. 14. The model is also shown below. In this model, the α -helical rods align parallel to each other to

form two-dimensional (2D) planar bilayer (membrane), followed by the transformation of the planar bilayer into the vesicle. It should be noted that the parallel alignment of α -helical rods in self-assembled vesicles have been reported in other rod-coils (*J. Am. Chem. Soc.* **2016**, *138*, 5773-5776). It is well-known that vesicles are formed by the transformation of 2D membranes. The formation of 2D membranes by the parallel alignment of rod-coils also supports our model (*Angew. Chem. Int. Ed.* **2009**, *48*, 1664-1668; *Angew. Chem. Int. Ed.* **2009**, *48*, 3657-3660).

Supplementary Figure 14. Schematic model of vesicles formed by the self-assembly of C- α_m -PEG₁₀ ($m = 3-6$).

As the reviewer commented, we added the DLS data of C- α_7 -PEG₁₀ as Supplementary Fig. 15. The diameter of micelles should be twice that of the molecular length of the building block. Because the average diameter (256 nm) of the nanostructures assembled from C- α_7 -PEG₁₀ is significantly larger than the molecular length of C- α_7 -PEG₁₀ (~ 2 nm), the spherical nanostructures observed in AFM should be vesicles rather than micelles.

Supplementary Figure 15. DLS data of C- α_7 -PEG₁₀.

We mentioned these explanations as follows in the revised manuscript: “When the m was from 3 to 6, vesicle-like spherical assemblies³¹⁻³³ were formed both without and with the magnetic field (Fig. 2f and Supplementary Figs. 14 and 15). However, the magnetic field-induced morphological transformation of molecular assemblies could be observed when the m was increased to 7 or 8.” (page 12, lines 19–22).

And, we added the following references in the revised manuscript:

31. Chen, X., He, Y., Kim, Y. & Lee, M. Reversible, Short alpha-Peptide Assembly for Controlled Capture and Selective Release of Enantiomers. *J. Am. Chem. Soc.* **138**, 5773-5776 (2016).
32. Hong, D. J. et al. Solid-state scrolls from hierarchical self-assembly of T-shaped rod-coil molecules. *Angew. Chem. Int. Ed.* **48**, 1664-1668 (2009).
33. Lee, E., Kim, J. K. & Lee, M. Reversible scrolling of two-dimensional sheets from the self-assembly of laterally grafted amphiphilic rods. *Angew. Chem. Int. Ed.* **48**, 3657-3660 (2009).

(Comment 3)

Without MF, C- α 5-PEG10 formed large rods, the authors should illustrate and explain the underlying mechanism.

(Response 3)

We understand that the reviewer expects us to explain the underlying mechanism of large rod formation after the MF treatment for C- α 7-PEG10. We added the pertinent model as Supplementary Fig. 16b in the revised manuscript.

Supplementary Figure 16. Effect of MF on the self-assembly of C- α 7-PEG10. **a**, AFM images before and after the MF treatment. **b**, Model of self-assembly processes in the absence or presence of the magnetic field.

We mentioned these explanations as follows in the revised manuscript: “However, the entropic penalty associated with self-assembly should be smaller for the aligned rod-coils than

for the rod-coils randomly tumbling in solution (Fig. 2h and Supplementary Fig. 16b). Thus, the aligned rod-coils will participate in the self-assembly process more readily than the others, which would alter the overall dynamics of supramolecular crystal growth.” (page 13, lines 1–2).

(Comment 4)

In Fig.2c, the authors designed and studied the properties of linear and circular amphiphiles, however, when studying assembly behaviors, they chose C- α 5-PEG10 and α 10-RGD3, why? The α helix lengths and the hydrophilic segments were different (5 vs 10, and PEG vs RGD3), I feel this part is very weak. The authors should provide more data to explain how α helix length and hydrophilic segments affect the assembly process.

(Response 4)

First of all, we would like to point out that the selection of C- α 7-PEG₁₀ (as mentioned in the response 2, we mentioned that C- α 5-PEG₁₀ is C- α 7-PEG₁₀) or α 10-RGD₃ was not random. In fact, before reaching to the conclusion that self-assembly process of C- α 7-PEG₁₀ and α 10-RGD₃ are responsive to MF, we have synthesized many more building blocks and have performed a lot of control experiments with those building blocks that was not illustrated in the original manuscript. During the preparation of the original manuscript, we did not fully explain the results from those control building blocks to avoid an increase in the length of the manuscript. Having commented about the building block selection issue from the reviewer, we now provide more data to explain how the α -helix length and the length of the hydrophilic segment affect the self-assembly process and the magnetic responsiveness.

As shown in Fig. 2f and Supplementary Fig. 18, we gradually increased the length of a hydrophobic α -helical segment of C- α _m-PEG₁₀ ($m = 3-8$) and L- α _m-PEG₁₀ ($m = 3-8$) while maintaining the length of the PEG segment. We found that the changes in the self-assembled morphology of C- α _m-PEG₁₀ could be observed after MF treatment when the m became more than 7 (i.e., C- α 7-PEG₁₀). In the case of the linear rod-coils, the magnetic responsiveness could be observed from $m = 4$. Thus, the magnetically responsive α -helical segment should of certain minimal length for the overall self-assembly process can be affected by MF treatment.

We mentioned these explanations as follows in the revised manuscript: “We investigated the effect of magnetic field on the self-assembly in a series of cyclic and linear rod-coils (C- α _m-PEG₁₀ and L- α _m-PEG₁₀, $m = 3-8$). We progressively increased the length of the hydrophobic α -helical rod block while maintaining the length of hydrophilic PEG block. The experiments were performed, at which the concentrations of the building blocks were higher than their critical aggregation concentrations (CACs) (Supplementary Fig. 13 and Supplementary Table 5). When the m was from 3 to 6, vesicle-like spherical assemblies³¹⁻³³ were formed both without and with the magnetic field (Fig. 2f and Supplementary Figs. 14 and 15). However, the magnetic field-induced morphological transformation of molecular assemblies could be observed when the m was increased to 7 or 8. Especially, the transformation was most pronounced when the m was 7

including the formation of cruciform anisotropic nanostructures, which are unlikely to be found in typical molecular assemblies (Fig. 2f and Supplementary Fig. 16). The level of transformation tends to be more highly affected as the strength of the magnetic field increased from 0.07 T to 0.25 T (Fig. 2f,g). Magnetically induced transformation could not be observed at a concentration below CAC (Supplementary Fig. 17). The data indicate that a certain minimal length of the α -helical rod is required for magnetically responsive self-assembly.

Helix length dependence of morphological transformation was also observed in the linear rod-coils, L- α_m -PEG₁₀ (Supplementary Fig. 18). The minimal length of the α -helical rod ($m = 4$) required for the transformation was shorter than that of the cyclic rod-coils. When the m was more than 6, the linear rod-coils formed irregular aggregates rather than forming discrete assemblies, possibly due to the strong propensity to make hydrophobic interactions.” (page 12, lines 14–23) & (page 13, lines 1–10).

Fig. 2f. Effect of MF (0.25 T) on the self-assembly of C- α_m -PEG₁₀ ($m = 3-8$). The peptide concentration was 20 μ M, which is higher than the CAC.

Supplementary Figure 18. Effect of MF (0.25 T) on the self-assembly of L- α_m -PEG₁₀ ($m = 3-8$).

In the case of α_{10} -RGD₃, we found that α_{10} -RGD₂ that has a shorter hydrophilic RGD unit was not affected by MF in its self-assembly behavior whilst the increase in the length of hydrophilic RGD unit in α_{10} -RGD₃ made it possible for the molecule to be affected by MF in its self-assembly behavior. Because the tendency to self-assemble increases as the hydrophobic/hydrophilic ratio increases, we interpret that α_{10} -RGD₂ was not responsive to MF because the molecule forms too strong molecular assembly to be affected by the MF strength used in the study. As described above, similar effect was also observed in L- α_m -PEG₁₀ ($m = 3-8$). Thus, the extent of morphological transformation after MF treatment is dependent on the balance between the magnetic responsiveness of the building block molecule and the hydrophobic/hydrophilic ratio.

We mentioned these explanations as follows in the revised manuscript: “Another type of linear rod-coil, α_{10} -(RGD)₂, was also tested for magnetic transformation; however, we could not observe magnetic transformation with this molecule. In comparison, the magnetic transformation into anisotropic and unique planar nanostructures even at the weak magnetic field strength (0.07 T) was evident when the length of a hydrophilic block was increased in α_{10} -(RGD)₃ (Supplementary Fig. 19). Together, the hydrophobic-hydrophilic balance, the length of the α -helical rod, and topology (cyclic or linear) is important for magnetically responsive self-assembly.” (page 13, lines 10–17).

(Comment 5)

The authors claimed that they fabricated the artificial chromosomes. What are the driven forces of the assembly of artificial chromosomes? Is it the electrostatic interactions or the hydrophobic interactions between α helices? Only two positive charges and the weak hydrophobic interactions seemed difficult to package DNA as the living system did.

(Response 5)

As mentioned in the manuscript, the main driving force for the self-assembly into artificial chromosomes are the electrostatic interactions between the peptide building block and DNA, and hydrophobic interactions between the peptides. These two forces work cooperatively with each other. The proper balance between the electrostatic and hydrophobic interactions and cooperativity should be crucial for the formation of the artificial chromosome. We found that R₁- α_{10} -PEG₁₆, in the absence of DNA, self-assembled into spherical nanostructures (Fig. 3c and Supplementary Fig. 21). The formation of the artificial chromosome was not efficient when we added DNA into the preformed spherical molecular assembly of R₁- α_{10} -PEG₁₆. Efficient formation of the artificial chromosome was possible only when R₁- α_{10} -PEG₁₆ was first disrupted by sonication before the injection into the solution DNA (please see the Experimental section). This is evidence that the R₁- α_{10} -PEG₁₆ and DNA associate together via molecular interactions when they exist in the molecularly dispersed state.

Supplementary Figure 21. Spherical nanostructures formed by the self-assembly of R_1 - α_{10} -PEG₁₆.

Additionally, we actually prepared building blocks that have more positive charges [R_2 - α_{10} -PEG₁₆ (3 positive charges), R_3 - α_{10} -PEG₁₆ (4 positive charges) and R_4 - α_{10} -PEG₁₆ (5 positive charges)] before deciding to use R_1 - α_{10} -PEG₁₆ (2 positive charges) for further studies. As added in Supplementary Fig. 22, the building blocks containing higher number of positive charges could make DNA complexes more efficiently at low +/- ratios. This is evidence that electrostatic interactions between the peptide and DNA are involved in the complex formation. R_1 - α_{10} -PEG₁₆ was selected for further studies among them. Because R_1 - α_{10} -PEG₁₆ has the weakest electrostatic attraction between the peptide and DNA, it would be more suitable for DNA release experiments.

Supplementary Figure 22. EMSA of the linear plasmid DNA with the building blocks having 3–5 positive charges. **a**, $R_2\text{-}\alpha_{10}\text{-PEG}_{16}$, **b**, $R_3\text{-}\alpha_{10}\text{-PEG}_{16}$, **c**, $R_4\text{-}\alpha_{10}\text{-PEG}_{16}$.

We mentioned these explanations as follows in the revised manuscript: “Given the usefulness of rod-coils in the formation of well-defined assemblies with DNA such as filamentous virus-like particles^{35,36}, we designed modified MNP-helix rod-coils ($R_x\text{-}\alpha_{10}\text{-PEG}_{16}$, $x = 1\text{--}4$), that have positively charged arginine (Arg or R) residues at the distal end of the hydrophobic α_{10} rod (Fig. 3a). The $\alpha_{10}\text{-PEG}_{16}$ was designed as a basis of the above building blocks considering the hydrophobic-hydrophilic balance.” (page 16, lines 15–18). “The gradual band retardation in EMSA as the charge ratio (+/-) increased is an indication of PD-complex formation (Fig. 3d and Supplementary Fig. 22). The band smearing is an indication of cooperatively assembled complex formation because multimers containing different number of building blocks can exist simultaneously in cooperative assemblies^{35,37,38}. The building blocks containing higher number

of positive charges could make DNA complexes more efficiently at low +/- ratios. The building block without positive charge, α_{10} -PEG₁₆, was not able to make PD-complexes (Supplementary Fig. 23). These are evidences that electrostatic interactions between the peptide and DNA are crucial for the complex formation. R₁- α_{10} -PEG₁₆ was selected for further in-depth studies among the four building blocks anticipating that weakly associated PD-complexes would be more magnetically responsive.” (page 17, lines 8–18).

(Comment 6)

The authors used R- α_{10} -PEG₁₆ to compact DNA, why chose α_{10} and PEG₁₆ ? The assembly and disassembly of solely R- α_{10} -PEG₁₆ should be studied as it is very important what kind of supramolecular structures it will form.

(Response 6)

For R_a- α_b -PEG_c type molecules that are to be used for DNA complexation, there can be three variables, i.e., *a*, *b*, and *c*. First, we fixed *b* = 10, i.e., α_{10} , as mentioned in the manuscript, is a perfect α -helix in the monomeric state (MNP-helix). By reducing the number of variables, the explanation of the DNA complexation behavior can be more simplified. Regarding the length of hydrophilic PEG unit (i.e., *c*), we found that water solubility of the building decreased when the PEG unit length was too short. If the length of the PEG unit is too long, the self-assembly tendency to the building block would be decreased. Our data and structural analysis indicated that PEG₁₆ was suitable to use in combination with α_{10} considering the hydrophilic and hydrophobic balance. Instead of making large number of analogues with different values of *c*, we decided to employ *a* (*a* = 1–4) as a variable, as described in the Response 5.

As the reviewer commented, we studied the assembly and disassembly of solely R₁- α_{10} -PEG₁₆ and the result is added as Supplementary Fig. 21 and 30. In the absence of MF, R₁- α_{10} -PEG₁₆ assembles into spherical nanostructures (Fig. 3c, Supplementary Fig. 21). After the exposure to RMF, the spherical nanostructures became more irregular (Supplementary Fig. 30). Thus, the molecular assemblies of R₁- α_{10} -PEG₁₆ themselves are also responsive to MF.

Supplementary Figure 30. Effect of RMF exposure on the self-assembled nanostructures from R- α_{10} -PEG₁₆.

We mentioned these explanations as follows in the revised manuscript: “The nanoassemblies of R- α_{10} -PEG₁₆ are also affected by RMF (Supplementary Fig. 30).” (page 23, lines 10–11).

(Comment 7)

Important controls, such as α_{10} -PEG₁₆ without positive charge also should be provided in Fig. 5

(Response 7)

As the reviewer pointed out, EMSA data of α_{10} -PEG₁₆, which has no positive charge, with the linear plasmid DNA was performed and the result from this control experiment is added as Supplementary Fig. 23. The result show that α_{10} -PEG₁₆ does not interact with the DNA at all. Thus, this result demonstrates that the positively charged parts (R_n) of R- α_{10} -PEG₁₆ are crucial for the complex formation.

Supplementary Figure 23. EMSA of the linear plasmid DNA with α_{10} -PEG₁₆.

We mentioned these explanations as follows in the revised manuscript: “The building block without positive charge, α_{10} -PEG₁₆, was not able to make PD-complexes (Supplementary Fig. 23). (page 17, lines 13–15).

(Comment 8)

Supplementary Fig. 2~3 should be mentioned in the main text.

(Response 8)

As suggested by the reviewer, we mentioned the Supplementary Fig. 2~3 in the main text as follows, “To find the minimal length necessary for MNP-helix formation, we progressively increased the length of the peptides composed of alternating combinations of Ala and Aib (α_m),

while the length of the hydrophilic coil segment (RGD) remained constant [α_m -(RGD)₂; Fig. 1c,d and Supplementary Figs. 1–3].” (page 7, lines 19–22).

Reviewer #2:

(Comment 1)

The authors report on exploiting the positive magnetic susceptibility of alpha helices to magnetically control assembly/disassembly of rod-coil molecules. They developed a library of rod-coil and coil-rod-coil molecules where the coil segment is either RGD or a PEG unit. CD was used to determine the helicity while AFM was used to determine the morphology of higher order structures. Finally, the authors designed one rod-coil monomer bearing a single Arginine to package DNA and demonstrate that under exposure to a rotating magnetic field they can disassemble these structures.

While I find the concept of utilizing the magnetic susceptibility of peptide motifs and self-assembly to potentially create functional structures interesting, I do not think this particular work demonstrates it well. The design space is limited, there is a lack of sufficient hypotheses, missing experimental evidence for the claims made, and the application is not exciting. There are lots of gaps to fill before this can be considered for publication in a high level and broad audience journal such as Nature Communications. To help the authors understand what is missing, I outlined detailed comments below:

(1) Molecular design of rod-coil monomers:

The biggest criticism I have relates to the rationale behind the monomer library and how these choices were made. In the current form of the manuscript, most choices seem random, scattered, with no hypothesis or sufficient data to support it. It also does not inform the scientific community on what is needed or is critical/important to design similar materials or enhance their magnetic susceptibility. Specific comments below:

(Response 1)

We would like to thank the reviewer for thoughtful and helpful comments on our manuscript. We would like to make responses to the reviewer's comments. Revised phrases and sentences are highlighted in blue in the manuscript and the supplementary information.

We would like to point out that, before deciding to show some of the results from selected building blocks, we already have synthesized many more building blocks and have performed a lot of control experiments with those building blocks that was not illustrated in the original manuscript. During the preparation of the original manuscript, we did not fully explain the results from those control building blocks to avoid an increase in manuscript length. Having commented about the lack of sufficient hypotheses and experimental evidence from the reviewer,

we now provide more data in the revised manuscript to explain why we selected some of the building blocks for the magnetic responsiveness and DNA packaging and how we were able to draw conclusions from the research findings. For more details, please see the responses below.

(Comment 2)

Most design choices are not explained or supported by sufficient data. The authors used alternating A-Aib - would another order work too? What design parameters are most important?

(Response 2)

As commented by the reviewer, we added more detailed explanation about the helix design. *We mentioned these explanations as follows in the revised manuscript: “Homopolymer of Aib was excluded from the design because multiple Aib repeats induced the formation of 3_{10} helices rather than α -helices. Thus, we anticipated that a certain combination of Ala and Aib in peptides might produce the MNP-helix with an appropriate hydrophobicity for self-assembly. We considered the possibility of using the repeats of Ala-Aib or Ala-Aib-Aib. Because we found that the synthetic yield dropped significantly for a contiguous stretch of Aib, we decided to employ the Ala-Aib repeat in designing MNP-helix.” (page 7, lines 11–17).*

(Comment 3)

Why RGD the coil domain and how was its length determined? - this choice seems random and is not explained or rationalized.

(Response 3)

At the initial phase of the research, we chose RGD because the sequence is hydrophilic. We also considered the further application potential because RGD sequence is known to bind to integrin receptors abundant in cancer cells. After we found that α_{10} is a perfect α -helix using several variants, i.e., α_m -(RGD)_n, we further confirmed the result using a simpler hydrophilic segment, i.e., PEG. The RGD length was designed by considering hydrophilic-hydrophobic balance and appropriateness of the RGD length was determined experimentally considering water solubility and self-assembly tendency. As commented by the reviewer, we added more detailed explanation about why used RGD coil and how its length determined.

We mentioned these explanations as follows in the revised manuscript: “Initially, the RGD peptide was selected as a hydrophilic block because it is a well-known functional peptide targeting integrin receptors highly expressed in cancer cells.²⁵” (page 8, lines 1–2). “Another type of linear rod-coil, α_{10} -(RGD)₂, was also tested for magnetic transformation; however, we could not observe magnetic transformation with this molecule. In comparison, the magnetic transformation into anisotropic and unique planar nanostructures even at the weak magnetic field strength (0.07 T) was evident when the length of a hydrophilic block was increased in α_{10} -(RGD)₃ (Supplementary Fig. 19).” (page 13, lines 10–15).

The reference 25 was added in the revised manuscript.

“25. Plow, E. F. et al. Ligand binding to integrins. *J. Biol. Chem.* **275**, 21785-21788 (2000).”

(Comment 4)

AFM is not sufficient to determine that monomers are not assembling. DLS is required to see that these are indeed monomers in the experimental conditions and don't form higher-order structures.

(Response 4)

As pointed out by the reviewer, the DLS measurements of PEG₃₀- α ₁₀-PEG₃₀ were conducted in aqueous solutions at various concentrations (Supplementary Fig. 7). Even at concentration as high as 1 mM, the particle size was measured to be smaller than 10 nm, which confirmed that the molecule did not form higher-order structures and existed as monomers.

Supplementary Figure 7. DLS data of PEG₃₀- α ₁₀-PEG₃₀ in aqueous solution **a**, 10 μ M. **b**, 100 μ M. **c**, 1 mM.

We mentioned this explanation as follows in the revised manuscript: “It was confirmed that PEG₃₀- α ₁₀-PEG₃₀ is also highly helical and monomeric (Supplementary Figs. 6–8).” (page 10, lines 20–21).

(Comment 5)

If the magnetic field can induce the peptide to form a stable alpha helix, it is not clear why starting with alpha10 versus alpha5 is better for all the other work presented in the paper.

(Response 5)

We guess that the meaning of the reviewer’s comment about “all the other work presented in the paper” is the study related to the complex formation with DNA. For R_a- α _b-PEG_c type molecules that are to be used for DNA complexation, there can be three variables, i.e., *a*, *b*, and *c*. First, we fixed *b* = 10, i.e., α ₁₀, as mentioned in the manuscript, is a perfect α -helix in the monomeric state (MNP-helix). By reducing the number of variables, the explanation of the DNA complexation behavior can be more simplified. The α ₅ is also helical but is not a perfect α -helix. For self-assembly in the absence of magnetic field such as in the DNA complexation experiment, α ₅ cannot function as a highly rigid rod in rod-coils. Please see the Response 12 below for more details.

(Comment 6)

From RGD on one end - coil was changed to a peg16 unit on both ends. Authors claim it is "to avoid self-assembly". Again - why peg16 versus any other peg length. Why capping the rod at both sides? Would one be sufficient and which one? Later monomers in the paper have only one peg arm - so this point seems important to demonstrate.

(Response 6)

The building block with the hydrophilic RGD segment was designed for some biological applications because RGD peptide is known to bind an integrin receptor, which is more abundantly expressed in cancer cells than normal cells. Biological application using RGD-integrin interactions can be the subject of further studies. The reasons why we changed the RGD to PEG is: first, after obtaining the initial results that there was a possibility that α ₁₀ could be the MNP-helix, we want to further verify the result with in-depth physico-chemical studies using a more simplified system (i.e., PEG) than the RGD. Furthermore, as mentioned in the manuscript, we needed to use organic solvents (methanol-d₃ and chloroform-d) for NMR characterization. Because PEG is soluble not only in water but also in many organic solvents such as chloroform, dichloromethane, and acetonitrile, the multi-solvent solubility of PEG was also a reason to use PEG for further studies.

We mentioned these explanations as follows in the revised manuscript: “To exclude the possibility of assembly-induced helix stabilization, we designed PEG₁₆- α ₁₀-PEG₁₆, which is

unlikely to self-assemble due to the presence of bulky hydrophilic blocks at both ends of the hydrophobic α_{10} unit. Here, PEG was used as a hydrophilic block for further in-depth physico-chemical studies using a more simplified molecule than the RGD peptide.” (page 8, lines 9–13).

Regarding the reason why we chose PEG16 versus any other PEG length, at first, we chose PEG16 in PEG₁₆- α_{10} -PEG₁₆ by considering a weight fraction (or volume fraction) between the hydrophilic segments (i.e., PEGs at both ends) and the hydrophobic segment (i.e., α_{10}). In the case of PEG₁₆- α_{10} -PEG₁₆, the M.W. of two PEG16 segments is 1,781 (= 890.5 × 2) and the M.W. of the α_{10} segment is 1,632.9. Because the α_{10} segment is not highly hydrophobic, we hypothesized that such a PEG length is likely to be appropriate in preventing the self-assembly of the building block. We also hypothesized that it would be more effective in preventing the self-assembly of building blocks by placing PEGs at both ends rather than placing the PEG only in one end. In the case of PEG₃₀- α_{10} -PEG₃₀, we increased the PEG length to improve the solubility of the building block in organic solvent such as chloroform and methanol because we found that PEG₁₆- α_{10} -PEG₁₆ has limited solubility in those organic solvents. The later monomer, R- α_{10} -PEG₁₆, was designed to have only one PEG arm because we needed to place a positively charged arginine (R) in the other end for the electrostatic interaction with negatively charged DNA.

Because mentioning of all these explanations would significantly lengthen the manuscript, we briefly mentioned the explanations in the manuscript as follows; “For NMR characterization, PEG₃₀- α_{10} -PEG₃₀ that has longer PEG chain compared to that of PEG₁₆- α_{10} -PEG₁₆, was designed to increase the solubility in organic solvent.” (page 10, lines 18–19).

(Comment 7)

The authors claim the peg units have no role in stabilizing the helix formation of the peptide, but this claim is not supported by data. In fact, peg molecules are known themselves to form helices and coils (Macromolecules 2005, 38, 9333-9340), so it's not clear how this effect can be decoupled.

(Response 7)

It should be noted that the molecular should be chiral in order to have any CD signal; but PEG is not chiral. Thus, PEG is not likely to have CD signal. In addition, the PEG, in the reference commented by the reviewer (Macromolecules 2005, 38, 9333-9340), was reported to form helix in pure isobutyric acid and isobutyric acid-rich aqueous solution. Thus, our solution condition (i.e., aqueous solution) is different from the isobutyric acid-rich solution condition known to induce PEG helices. Moreover, the PEG helix is structurally different from α -helix.

To make it sure that PEG does not have any CD signal (thus, any CD signal from the rod-coil building blocks arises only from peptide segments) in our experimental condition, we investigated PEG1000 that has comparable M.W. as that of PEG16 with CD spectroscopy. As shown in Supplementary Fig. 9, PEG1000 does not have any CD signal at all.

Supplementary Figure 5. Comparison of CD spectra between PEG1000 and PEG₁₆-α₁₀-PEG₁₆ at the identical experimental condition. (50 μM in water).

We mentioned these explanations as follows in the revised manuscript: “It should be noted that the PEG block does not have any CD signal (Supplementary Fig. 5).” (page 8, lines 14–15).

(Comment 8)

Next, for NMR it was changed to Peg30... but no CD or AFM characterization was done for this one or compared to the peg16 one or explanation provided why peg16 could not be explored with NMR!

(Response 8)

As commented by the reviewer, we added the CD and AFM data of PEG₃₀-α₁₀-PEG₃₀ as Supplementary Figs. 6 and 8a. The data show that PEG₃₀-α₁₀-PEG₃₀ exists as monomeric helix similar to PEG₁₆-α₁₀-PEG₁₆.

As mentioned in the Response 6, we changed PEG16 to PEG30 because PEG₁₆-α₁₀-PEG₁₆ has limited solubility in the organic solvent (the mixture of chloroform and methanol) used for NMR study. Please see the Response 6 for more details.

Supplementary Figure 6. CD data of PEG₃₀- α ₁₀-PEG₃₀. **a**, CD spectra obtained at various concentrations. **b**, Concentration independence of the helicity.

Supplementary Figure 8. AFM images of PEG₃₀- α ₁₀-PEG₃₀ at 1 mM with or without magnetic field. **a**, The sample dried on mica without MF. **b**, The sample dried on mica in presence of static MF.

We mentioned these explanations as follows in the revised manuscript: “For NMR characterization, PEG₃₀- α ₁₀-PEG₃₀ that has longer PEG chain compared to that of PEG₁₆- α ₁₀-PEG₁₆, was designed to increase the solubility in organic solvent. It was confirmed that PEG₃₀- α ₁₀-PEG₃₀ is also highly helical and monomeric (Supplementary Figs. 6–8).” (page 10, lines 18–21).

(Comment 9)

Then, for the self-assembly - they explored linear and cyclic rod-coils - but again without any rational or hypothesis driving this design choice. And once again - peg unit length was changed to peg10 - no explanation!!

(Response 9)

As commented by the reviewer, we provide the rationale behind the design of the linear and cyclic rod-coils in the revised manuscript.

We mentioned these explanations as follows in the revised manuscript: “We investigated the effect of magnetic field on the self-assembly in a series of cyclic and linear rod-coils (C- α _m-PEG₁₀ and L- α _m-PEG₁₀, $m = 3–8$). We progressively increased the length of the hydrophobic α -helical rod block while maintaining the length of hydrophilic PEG block.” (page 12, lines 14–17). “Helix length dependence of morphological transformation was also observed in the linear rod-coils, L- α _m-PEG₁₀ (Supplementary Fig. 18).” (page 13, lines 6–7).

(Comment 10)

Self-assembly changes were only explored for Cyclic- α_5 -peg10 and linear- α_{10} -(RGD)₃ - why these ones, what happens for the others, and why? Can this behavior be predicted or generalized?

Linear- α_{10} -(RGD)₃ seems particularly random - why coming back to RGD again? and this time with 3 RGD repeats? Why does this one behave differently than any other α -RGD monomer in the library?

(Response 10)

We would like to make a correction that C- α_5 -PEG₁₀ mentioned in the original manuscript was C- α_7 -PEG₁₀. We correct this error in the revised manuscript. Appropriate data related to characterization of C- α_5 -PEG₁₀ are supplemented in the revised manuscript (see below).

First of all, we would like to point out that the selection of C- α_7 -PEG₁₀ and α_{10} -RGD₃ was not random. In fact, before reaching to the conclusion that self-assembly process of C- α_7 -PEG₁₀ and α_{10} -RGD₃ are responsive to MF, we have synthesized many more building blocks and have performed a lot of control experiments with those building blocks that was not illustrated in the original manuscript. During the preparation of original manuscript, we did not fully explain the results from those control building blocks to avoid an increase in the manuscript length. Having commented about the building block selection issue from the reviewer, we now provide more data to explain how α -helix length and the length of the hydrophilic segment affect the self-assembly process and magnetic responsiveness.

As shown in Fig. 2f, we increased the length of a hydrophobic α -helical segment of C- α_m -PEG₁₀ ($m = 3-8$) while maintaining the length of the PEG segment. We found that the changes in the self-assembled morphology of C- α_m -PEG₁₀ could be observed after MF treatment when the m became more than 7 (i.e., C- α_7 -PEG₁₀). Thus, the magnetically responsive α -helical segment should of certain minimal length for the overall self-assembly process can be affected by MF treatment. Dependence of magnetic responsiveness on the length of the α -helical segment was also observed in the linear building block, i.e., L- α_m -PEG₁₀ ($m = 3-8$). This data is added as Supplementary Fig. 18.

Fig. 2f. Effect of MF (0.25 T) on the self-assembly of C- α_m -PEG₁₀ ($m = 3-8$). The peptide concentration was 20 μ M, which is higher than the CAC.

Supplementary Figure 18. Effect of MF (0.25 T) on the self-assembly of L- α_m -PEG₁₀ ($m = 3-8$).

We mentioned these explanations as follows in the revised manuscript: “We investigated the effect of magnetic field on the self-assembly in a series of cyclic and linear rod-coils (C- α_m -PEG₁₀ and L- α_m -PEG₁₀, $m = 3-8$). We progressively increased the length of the hydrophobic α -helical rod block while maintaining the length of hydrophilic PEG block. The experiments were performed, at which the concentrations of the building blocks were higher than their critical aggregation concentrations (CACs) (Supplementary Fig. 13 and Supplementary Table 5). When the m was from 3 to 6, vesicle-like spherical assemblies³¹⁻³³ were formed both without and with the magnetic field (Fig. 2f and Supplementary Figs. 14 and 15). However, the magnetic field-induced morphological transformation of molecular assemblies could be observed when the m was increased to 7 or 8. Especially, the transformation was most pronounced when the m was 7 including the formation of cruciform anisotropic nanostructures, which are unlikely to be found in typical molecular assemblies (Fig. 2f and Supplementary Fig. 16). The level of transformation tends to be more highly affected as the strength of the magnetic field increased from 0.07 T to 0.25 T (Fig. 2f,g). Magnetically induced transformation could not be observed at a concentration below CAC (Supplementary Fig. 17). The data indicate that a certain minimal length of the α -helical rod is required for magnetically responsive self-assembly.

Helix length dependence of morphological transformation was also observed in the linear rod-coils, L- α_m -PEG₁₀ (Supplementary Fig. 18). The minimal length of the α -helical rod ($m = 4$) required for the transformation was shorter than that of the cyclic rod-coils. When the m was more than 6, the linear rod-coils formed irregular aggregates rather than forming discrete assemblies, possibly due to the strong propensity to make hydrophobic interactions.” (page 12, lines 14–23) & (page 13, lines 1–10).

In the case of α_{10} -RGD₃, we found that α_{10} -RGD₂ that has a shorter hydrophilic RGD unit was not affected by MF in its self-assembly behavior whilst the increase in the length of hydrophilic RGD unit in α_{10} -RGD₃ made it possible for the molecule to be affected by MF in its self-assembly behavior. Because the tendency to self-assemble increases as the hydrophobic/hydrophilic ratio increases, we interpret that α_{10} -RGD₂ was not responsive to MF because the molecule forms too strong molecular assembly to be affected by the MF strength used in the study. Thus, the extent of morphological transformation after MF treatment is dependent on the balance between the magnetic responsiveness of the building block molecule (the length of the α -helical segment) and the hydrophobic/hydrophilic ratio.

We mentioned these explanations as follows in the revised manuscript: “Another type of linear rod-coil, α_{10} -(RGD)₂, was also tested for magnetic transformation; however, we could not observe magnetic transformation with this molecule. In comparison, the magnetic transformation into anisotropic and unique planar nanostructures even at the weak magnetic field strength (0.07 T) was evident when the length of a hydrophilic block was increased in α_{10} -(RGD)₃ (Supplementary Fig. 19). Together, the hydrophobic-hydrophilic balance, the length of the α -helical rod, and topology (cyclic or linear) is important for magnetically responsive self-assembly.” (page 13, lines 10–17).

(Comment 11)

Why C- α_7 -peg10 needs 0.25T and linear- α_{10} -(RGD)₃ requires 0.07T for the self-assembly switch?

(Response 11)

Indeed, the morphology of C- α_7 -PEG₁₀ also transformed at 0.07 T; however, the extent of transformation was smaller than that at 0.25 T. To illustrate the dependence of morphological transformation on the MF strength, we added Fig. 2g in the revised manuscript.

Fig. 2g, Effect of MF strength on self-assembly of C- α_7 -PEG₁₀. [C- α_7 -PEG₁₀] = 20 μ M.

Overall, the effect of MF strength on self-assembly of C- α_7 -PEG₁₀ is as follows;

In the case of α_{10} -(RGD)₃, as described in the Response 10, α_{10} -(RGD)₃ was designed because α_{10} -RGD₂ was not responsive to MF possibly due to α_{10} -RGD₂'s strong self-assembling propensity. Because α_{10} -(RGD)₃ underwent significant morphological transformation even at 0.07 T, we provided the data at 0.07 T (this data is moved to Supplementary Information as Supplementary Fig. 19 in the revised manuscript).

We mentioned these explanations as follows in the revised manuscript: “The level of transformation tends to be more highly affected as the strength of the magnetic field increased from 0.07 T to 0.25 T (Fig. 2f,g).” (page 13, lines 2–3).

(Comment 12)

For the application - the authors chose Arg-alpha10-PEG16 - why this one versus all the other monomers they tested? Why only one Arginine? Comparison between linear and cyclic monomers will be interesting to see if package the DNA differently.

(Response 12)

For R_a- α_b -PEG_c type molecules that are to be used for DNA complexation, there can be three variables, i.e., *a*, *b*, and *c*. First, we fixed *b* = 10, i.e., α_{10} , as mentioned in the manuscript, is a perfect α -helix in the monomeric state (MNP-helix). By reducing the number of variables, the explanation of the DNA complexation behavior can be more simplified. Regarding the length of hydrophilic PEG unit, we found that water solubility of the building decreased when the PEG unit length was reduced. If the length of the PEG unit is too long, the self-assembly tendency to the building block would be decreased. Our data and structural analysis indicated that PEG16 was suitable to use in combination with α_{10} considering the hydrophilic and hydrophobic balance. Instead of making large number of analogues with different values of *c*, we decided to employ *a* (*a* = 1–4) as a variable.

In fact, we prepared building blocks that have more positive charges [R₂- α_{10} -PEG₁₆ (3 positive charges), R₃- α_{10} -PEG₁₆ (4 positive charges) and R₄- α_{10} -PEG₁₆ (5 positive charges)] before deciding to use R₁- α_{10} -PEG₁₆ (2 positive charges) for further studies. As added in Supplementary Fig. 22, the building blocks containing higher number of positive charges could

make DNA complexes more efficiently at low +/- ratios. This is evidence that electrostatic interactions between the peptide and DNA are involved in the complex formation. $R_1\text{-}\alpha_{10}\text{-PEG}_{16}$ was selected for further in-depth studies among them. Because $R_1\text{-}\alpha_{10}\text{-PEG}_{16}$ has the weakest electrostatic attraction between the peptide and DNA, it would be more suitable for DNA release experiments.

Supplementary Figure 22. EMSA of the linear plasmid DNA with the building blocks having 3–5 positive charges. **a**, $R_2\text{-}\alpha_{10}\text{-PEG}_{16}$, **b**, $R_3\text{-}\alpha_{10}\text{-PEG}_{16}$, **c**, $R_4\text{-}\alpha_{10}\text{-PEG}_{16}$.

We mentioned these explanations as follows in the revised manuscript: “Given the usefulness of rod-coils in the formation of well-defined assemblies with DNA such as filamentous virus-like particles^{36,37}, we designed modified MNP-helix rod-coils ($R_x\text{-}\alpha_{10}\text{-PEG}_{16}$, $x = 1\text{--}4$), that have positively charged arginine (Arg or R) residues at the distal end of the hydrophobic α_{10} rod (Fig. 3a). The $\alpha_{10}\text{-PEG}_{16}$ was designed as a basis of the above building blocks considering the

hydrophobic-hydrophilic balance.” (page 16, lines 15–19). “We then added increasing concentrations of $R_{\alpha 10}$ -PEG₁₆ to a fixed amount of the linear plasmid DNA, and the formation of the peptide/DNA complex (PD-complex) was monitored using an electrophoretic mobility shift assay (EMSA). The gradual band retardation in EMSA as the charge ratio (+/–) increased is an indication of PD-complex formation (Fig. 3d and Supplementary Fig. 22). The band smearing is an indication of cooperatively assembled complex formation because multimers containing different number of building blocks can exist simultaneously in cooperative assemblies^{36,38,39}. The building blocks containing higher number of positive charges could make DNA complexes more efficiently at low +/- ratios.” (page 17, lines 6–13).

As the reviewer’s comment, it should be interesting to study the differences in DNA packaging between the linear and cyclic monomers. Although it was not covered in this manuscript because it requires a lot of optimization process through the comparison of many more designs, it should be the subject of further ongoing studies.

(Comment 13)

(2) Assembly/Disassembly

The authors seem to "fish" for finding an interesting effect instead of being able to design for one. There is also insufficient evidence how (it at all) it is the magnetic field that is responsible for the effect seen. Specific comments below:

There are multiple assembly goals that should be investigated separately and explored via multiple techniques. One is unfolded helix to helix using magnetic field - the authors briefly show that for cycle alpha 5 - the magnetic field seems to be inducing the incomplete helix into a closely perfect one - this is an interesting effect that should be explored further.

(Response 13)

As the reviewer commented, it should be certainly interesting to perform further in-depth studies regarding the transformation from a unfolded helix to a folded helix using MF, possibly using multiple techniques and experimental settings. We believe that this is beyond the scope of this paper and should be the goal of further ongoing studies.

(Comment 14)

Second is the monomer to assembled state - I did not see results demonstrating the magnetic field can induce self-assembly of non-assembled monomers - this will be of high value to achieve.

(Response 14)

As the reviewer suggested, we tested whether MF could induce the self-assembly of the unassembled monomer of PEG₃₀- α_{10} -PEG₃₀. As shown in Supplementary Fig. 8, we found that there are some changes after MF exposure. Although discrete molecules assemblies were not

formed after the MF exposure, it is likely that the molecules are affected by MF and move in certain directions on the surface of mica.

Supplementary Figure 8. AFM images of PEG₃₀-α₁₀-PEG₃₀ at 1 mM with or without magnetic field. **a**, The sample dried on mica without MF. **b**, The sample dried on mica in presence of static MF. Although discrete molecules assemblies were not formed after the MF exposure to the monomeric building block molecule, i.e., PEG₃₀-α₁₀-PEG₃₀, it is likely that the molecules are affected by MF and move in certain directions on the surface of mica.

(Comment 15)

Third is the change of assembly from one morphology to the other using a magnetic field - the authors showed some evidence that this might be possible - but only for 2 of their monomers, and their sequences seem random. Authors should expand their investigation into additional monomers and explore how each domain (rod versus coil), (linear versus cyclic), experimental conditions, etc. affect this behavior. Also – authors will benefit from writing their manuscript with this above rationale of how I explained the various assembly behaviors explored in this manuscript for better clarity.

(Response 15)

As commented by the reviewer, we performed more in-depth studies using a series of designed building blocks (please also see the Response 10). From that study, we were able to generalize the findings in magnetic responsiveness.

As shown in Fig. 2f, we increased the length of a hydrophobic α-helical segment of C-α_m-PEG₁₀ ($m = 3-8$) while maintaining the length of the PEG segment. We found that the changes in the self-assembled morphology of C-α_m-PEG₁₀ could be observed after MF treatment when the m became more than 7 (i.e., C-α₇-PEG₁₀). Thus, the magnetically responsive α-helical segment should of certain minimal length for the overall self-assembly process can be affected by MF treatment. As shown in Supplementary Fig. 18, we also investigated the length dependence of the hydrophobic segment in L-α_m-PEG₁₀ ($m = 3-8$) while maintaining the length of the PEG

segment. We found that the changes in the self-assembled morphology of L- α_m -PEG₁₀ could be observed after MF treatment when the m became more than 4 (i.e., L- α_4 -PEG₁₀). Thus, the MF-induced morphological transformation was more pronounced with the linear building blocks than the cyclic building blocks. (please see the Response 10 for more details)

(Comment 16)

(2.2) The magnetic field effect on self-assembly should be explored across a range of concentrations of the monomers. CMC/CAC should be established for each monomer to determine the concentration of assembly and magnetic effect on assembly should be explored below and above the cmc.

(Response 16)

As the reviewer commented, we measured CACs for the linear and cyclic building blocks and the data are added as Supplementary Fig. 13 and Supplementary table 5. To compare the effect of concentration, we compared the magnetic responsiveness of C- α_7 -PEG₁₀ at the concentration above or below CAC. Supplementary Fig. 16a and Supplementary Fig. 17 are the data at the concentration above or below CAC, respectively. Below CAC, there was no significant difference in morphology in the absence or in the presence of MF treatment, possibly because the molecule could not form discrete assemblies at a concentration below CAC. Only irregular aggregates were observed maybe because of condensation effect caused by drying process during the preparing AFM sample on mica.

Supplementary Table 5. Critical aggregation concentrations (CACs).

Peptide	CAC ^a (μM)
L- α_3 -PEG ₁₀	1.56
L- α_4 -PEG ₁₀	1.58
L- α_5 -PEG ₁₀	1.95
L- α_6 -PEG ₁₀	2.01
L- α_7 -PEG ₁₀	1.98
L- α_8 -PEG ₁₀	n.d
C- α_3 -PEG ₁₀	1.55
C- α_4 -PEG ₁₀	1.60
C- α_5 -PEG ₁₀	1.97
C- α_6 -PEG ₁₀	2.03

C- α_7 -PEG ₁₀	1.96
C- α_8 -PEG ₁₀	1.95

^a CACs were determined by the concentration dependent changes in tryptophan fluorescence emission.¹ ^b n.d = not determined due to the limited solubility in aqueous solution.

Supplementary Figure 16. Effect of MF on the self-assembly of C- α_7 -PEG₁₀. **a**, AFM images before and after the MF treatment.

Supplementary Figure 17. Magnetic effect on the self-assembly of C- α_7 -PEG₁₀ at a concentration below CAC. AFM images. [C- α_7 -PEG₁₀] = 156 nM.

We mentioned these explanations as follows in the revised manuscript: “The experiments were performed, at which the concentrations of the building blocks were higher than their critical aggregation concentrations (CACs) (Supplementary Fig. 13 and Supplementary Table 5).” (page 12, lines 17–19). “Magnetically induced transformation could not be observed at a concentration below CAC (Supplementary Fig. 17).” (page 13, lines 3–4).

(Comment 17)

(3) Application

Current application is not particularly exciting and its performance is not convincing. Specific comment below:

(3.1) Again - choice behind why using R-Alpha10-Peg16 is not clear - see point 1 above.

(Response 17)

The signs of $\Delta\chi$ for α -helix and DNA are opposite each other (i.e., α -helix, $\Delta\chi > 0$; DNA, $\Delta\chi < 0$). Thus, we believe that it should be useful to fabricate the supramolecular complexes of the MNP-helix rod-coil and DNA. Because α -helix and DNA respond to the magnetic field in different directions, we hypothesized that the overall magnetic responsiveness would be significantly increased in the supramolecular complexes of R_x - α_{10} -PEG₁₆ and DNA.

We mentioned these explanations as follows in the revised manuscript: “Duplex DNA has a large magnetic anisotropy value as an organic molecule due to the presence of base stacking.^{28,30,31} DNA orients perpendicular to the external magnetic field because $\Delta\chi$ is negative for DNA.²⁹ Because the signs of $\Delta\chi$ for α -helix and DNA are opposite each other, it should be interesting to fabricate the supramolecular complexes of the MNP-helix rod-coil and DNA. Given the usefulness of rod-coils in the formation of well-defined assemblies with DNA such as filamentous virus-like particles^{36,37}, we designed modified MNP-helix rod-coils (R_x - α_{10} -PEG₁₆, $x = 1-4$), that have positively charged arginine (Arg or R) residues at the distal end of the hydrophobic α_{10} rod (Fig. 3a). The α_{10} -PEG₁₆ was designed as a basis of the above building blocks considering the hydrophobic-hydrophilic balance. Because α -helix and DNA respond to the magnetic field in different directions, we hypothesized that the overall magnetic responsiveness would be increased in the supramolecular complexes of R_x - α_{10} -PEG₁₆ and DNA.” (page 16, lines 11–22). “This study lays foundation to magnetically interface organic materials with magnetic devices and instruments, with application potentials in magnetically responsive bionanomaterials, molecular magnetic devices, and smart peptide/nucleic acid complexes highly responsive to magnetic field.” (page 24, lines 10–13).

As described in the Response 12, we actually prepared building blocks that have more positive charges [R_2 - α_{10} -PEG₁₆ (3 positive charges), R_3 - α_{10} -PEG₁₆ (4 positive charges) and R_4 - α_{10} -PEG₁₆ (5 positive charges)] before deciding to use R_1 - α_{10} -PEG₁₆ (2 positive charges) for further studies. As added in Supplementary Fig. 22, the building blocks containing higher number of positive charges could make DNA complexes more efficiently. This is evidence that electrostatic interactions between the peptide and DNA are involved in the complex formation. R_1 - α_{10} -PEG₁₆ was selected for further studies because the molecule has the weakest electrostatic attraction between the peptide and DNA, it would be more suitable for DNA release experiments.

(Comment 18)

(3.2) Stability against degradation was only monitored for 600 seconds - this is not exciting for a real world application and should be evaluated for at least several hours to be meaningful.

(Response 18)

According to the reviewer's comment, we confirmed that the PD-complex was stable against the DNase-mediated degradation for several hours and the result is added as Supplementary Fig. 28 in the revised Supplementary Information.

Supplementary Figure 28. Time-dependent DNase protection assay for the PD-complex.

We mentioned these explanations as follows in the revised manuscript: “A time-dependent DNase protection assay quantitatively demonstrated that DNA, in its naked state, degraded rapidly upon the addition of DNase I; however, DNA, when it was incorporated into the artificial chromosome, was completely protected from nuclease digestion during the prolonged incubation period (Fig. 4c and Supplementary Fig. 28).” (page 21, lines 15–18).

(Comment 19)

(3.3) Based on point 3.2 - 2 weeks were required for disassembly - how can we know if degradation of either DNA or peptide are not the driver for this?

(3.4) Even with the rotating magnetic disassembly - no evidence that DNA and peptides remained intact or that they can be repackaged in the absence of a magnetic field.

(3.5) No evidence that DNA is indeed separate from peptides and can be fully functional.

(Response 19)

To confirm that the degradation of either DNA or peptide are not the driver of disassembly after 2 weeks of incubation, we performed control studies, and the results are added as Supplementary Fig. 31. As shown in Supplementary Fig. 31a, the DNA remained intact after 2 weeks of incubation or 1 h of RMF treatment. The peptide also remained intact after 2 weeks of incubation or 1 h of RMF treatment (Supplementary Fig. 31b). We found that the disassembled DNA and peptide were not repackaged into the complex in the absence of MF after one day of incubation. As shown in Fig. 5g, the presence of a single DNA strand is evidence that the DNA is indeed separate from the peptides. In the observed multiple DNA strands, it is likely that several peptide molecules are still attached to the DNA (Fig. 5d, f, and h). Because the DNA remained intact as shown in Supplementary Fig. 31a, the DNA should be fully functional. Moreover, the sequence-

specific amplification of DNA from the PD-complex (artificial chromosome) using polymerase chain reaction (PCR) further verify that the sequence information of the DNA, which determines the functionality of DNA, remained unchanged (Fig. 4d).

Supplementary Figure 31. Intactness of the linear plasmid DNA and the peptide (R- α_{10} -PEG₁₆) after incubation. **a**, The integrity of the DNA after incubation. Lane 3: SDS was used to release the DNA from the PD-complex. Lanes 4 and 5: PD-complexes were incubated for 2 weeks. **b**, MALDI-TOF MS spectra. Intactness of the peptide after 2 weeks of incubation (left) and 1 h of RMF treatment (right).

We mentioned these explanations as follows in the revised manuscript: “It should be noted that the peptide and DNA remained intact after the exposure to RMF (Supplementary Fig. 31).” (page 23, lines 11–13).

(Comment 20)

Packaging of DNA under a magnetic field should be done to see if it affects the process.

(Response 20)

Following the reviewer’s comment, we tested whether the DNA can be packaged under MF. As shown in Supplementary. Fig. 32, the shift in electrophoretic mobility of the DNA could be observed both in the absence or presence of MF. The results imply that the peptide and DNA can make molecular interactions with each other even in the presence of MF. The morphologies of

the complexes at the MF-treated conditions (i.e., in the presence of static MF or RMF) are likely to differ from that in the absence of MF; however, we believe that such in-depth studies would be more appropriate as further research endeavors.

Supplementary Figure 32. Influence of MF on the complex formation between the peptide (R- α_{10} -PEG₁₆) and the linear plasmid DNA. Lane 1: DNA ladder, lane 2: the plasmid DNA, lane 3: Mixture of the peptide and the DNA, lane 4: Mixture of the peptide and the DNA in the presence of static MF (0.25 T, 1 h), lane 5: Mixture of the peptide and the DNA in the presence of RMF (0.25 T, 1,200 rpm, 1 h)

Reviewer #3:

(Comment 1)

The manuscript by Lim and coworkers describes the magnetic-field induced assembly of especially designed helical amphipathic peptides, also in combination with DNA, where they result in assemblies as “artificial chromosomes”, which can also be disassembled in turn, most efficiently by rotating magnetic fields. The conclusions in the manuscript are supported mainly by AFM and TEM, but also NMR spectroscopy on the monomeric peptides as proof for their helical nature. With respect to the latter, a chemical shift analysis could support the alpha-helical nature of the monomeric peptide in addition to RDCs. Also, only 21 resonances are observed in the NMR spectrum; however, the chemical structure shows several more NH pairs. Why are they not observed, and what are the assignments of the 21 peaks? Or at least peak groups?

(Response 1)

We would like to thank the reviewer for thoughtful and helpful comments on our manuscript. We would like to make responses to the reviewer's comments. Revised phrases and sentences are highlighted in blue in the manuscript and the supplementary information.

As commented by the reviewer, we assigned peak groups and marked the peak groups in ^1H - ^{15}N IPAP-HSQC NMR spectrum in the revised Fig. 2a. The reason why there are more than 21 peaks is because there are amide groups in the PEG30 (3 PEG molecules are connected via amide groups to make PEG30) as shown in Supplementary Fig. 1.

Figure 2: Behaviours of the MNP-helix rod-coils under a magnetic field. a, The amide region of the 900 MHz ^1H - ^{15}N IPAP-HSQC NMR spectrum of PEG₃₀- α ₁₀-PEG₃₀ in methanol- d_3 /chloroform- d (1:1) at 298 K, showing the alignment of the MNP-helix rod with the direction of the applied magnetic field. PEG: amide groups in PEG30 (see the Supplementary Fig. 1).

(Comment 2)

Other remarks:

p19 line 8: there does not seem to be DNA condensation in this experiment (as with histones or protamines), rather only DNA wrapping? So the argument that this system shows very high DNA packaging density does not hold? Or please make a better argument to show that the DNA is condensed (volume in isolation, volume with amphipathic peptide assemblies?).

(Response 2)

As the reviewer pointed out, we described a better argument about the DNA condensation in the artificial chromosome in the revised manuscript.

We mentioned these explanations as follows in the revised manuscript: “In chromosomes, elongated and fluttering DNA chains are hierarchically condensed into compact structures in which DNA is wrapped around histone proteins. DNA is also highly condensed into the compact structures in the artificial chromosome; however, the formation mechanism is different from that of biological chromosomes because DNA strands lie parallel to each other in the artificial chromosomes.” (page 24, lines 6–10).

(Comment 3)

p19 line 20: it is surprising that a standard PCR protocol allows to amplify this protected plasmid – which chemical added, or other step in the protocol, could allow this? Or was the DNA disincorporated before?

(Response 3)

Considering the reviewer’s comment, we provide a more detailed description of PCR protocol.

The PCR protocol in the Experimental section has been revised as follows; “To assess the capability of recovering the DNA information stored in an artificial chromosome, a polymerase chain reaction (PCR) was conducted to amplify a specific DNA segment of the linear plasmid DNA. The specific 310 bp segment was amplified using the sense primer 5’-ACG GAG ACT GGA GTC GAA GAG G-3’ and the antisense primer 5’-GTA GGG CAA CTA GTG CAT CTC CC-3’. Each reaction mixture (50 μ L) contained 50 ng of DNA, 10 pmol of each primer, and 25 μ L of 2 \times Quick Taq HS DyeMix (Toyobo, Japan). PCR amplification (30 cycles) was performed with a T100 Thermal Cycler (Bio-Rad, USA). Each cycle included an initial denaturation step at 94 $^{\circ}$ C for 2 min, a denaturation step at 94 $^{\circ}$ C for 30 sec, a primer annealing step at 58 $^{\circ}$ C for 30 sec, and a extension step at 68 $^{\circ}$ C for 1 min. The T_m of the sense primer was 65.9 $^{\circ}$ C and the T_m of the antisense primer was 66.4 $^{\circ}$ C. After the 30 cycles of amplification, the final extension step was conducted at 68 $^{\circ}$ C for 5 min. Then the samples were stored at 4 $^{\circ}$ C until use. The amplified products were investigated by agarose gel electrophoresis. The electrophoresis was conducted for 120 min at 80 V in a 2 % agarose gel using 1 \times TBE buffer. SYBR Safe DNA Gel Stain (Invitrogen, USA) was used for staining.”

(Comment 4)

p20 line 9: why is it expected that the assemblies disassemble at all in a magnetic field, which initially is used to induce them?

(Response 4)

We interpret that the peptide/DNA complex (PD-complex) can disassemble in a magnetic field because the signs of $\Delta\chi$ for α -helix and DNA are opposite each other. Thus, α -helix and DNA respond to the magnetic field in different directions facilitating the disassembly of the PD-complex, especially in the presence of rotating magnetic field. We modified Fig. 5i to illustrate the above-described effect.

Fig. 5i, Disassembly of the artificial chromosome by the rotating magnetic field.

We mentioned these explanations as follows in the revised manuscript: “Duplex DNA has a large magnetic anisotropy value as an organic molecule due to the presence of base stacking.^{28,30,31} DNA orients perpendicular to the external magnetic field because $\Delta\chi$ is negative for DNA.²⁹ Because the signs of $\Delta\chi$ for α -helix and DNA are opposite each other, it should be interesting to fabricate the supramolecular complexes of the MNP-helix rod-coil and DNA.” (page 16, lines 11–15). “Because α -helix and DNA respond to the magnetic field in different directions, we hypothesized that the overall magnetic responsiveness would be increased in the supramolecular complexes of R_x - α_{10} -PEG₁₆ and DNA.” (page 16, lines 19–22). “The opposite signs of $\Delta\chi$ between α -helix and DNA would have facilitated the magnetically induced unpacking of DNA from the artificial chromosomes (Fig. 5i).” (page 23, lines 9–10).

(Comment 5)

In general, the conclusions are difficult to understand for a non-specialist reader. For instance “Given the usefulness of rod-coils in the formation of well-defined assemblies with... “ – useful for what? Or: “The band smearing that occurred at a charge ratio of 1 was an indication of cooperatively assembled complex formation^{32,34,35}. “ – here should be shortly explained how these references support this conclusion. There are other specialist shortcuts in the manuscript, and an effort to better discuss the findings in the light of the references given would be in general useful for the reader.

(Response 5)

As the reviewer commented, we added the following explanations for general readers;

“Given the usefulness of rod-coils in the formation of well-defined assemblies with DNA such as filamentous virus-like particles^{32,33}, we designed modified MNP-helix rod-coils (R_x - α_{10} -PEG₁₆, $x = 1-4$), that have positively charged arginine (Arg or R) residues at the distal end of the hydrophobic α_{10} rod (Fig. 3a).” (page 16, lines 15–18). “The band smearing is an indication of cooperatively assembled complex formation because multimers containing different number of building blocks can exist simultaneously in cooperative assemblies^{36,38,39}.” (page 17, lines 10–12).

Reviewer #4:

The work of Lim et al., reports on the synthesis of alpha-helix-peptides-PEG block-copolymers with enhanced respond to magnetic fields. This offers the possibility to tune self-assembly processes remotely by the application of an external magnetic field. The system developed by these authors has the ability to respond to mild external magnetic fields which make applications more accessible. Additionally authors tested a selected alpha-peptide-PEG copolymer by its capacity to interact electrostatically with a DNA strand to form a supramolecular structure that is able to protect the DNA for enzymatic degradation and simultaneously do not avoid its replication. Moreover the peptide-PEG covering can be removed by the application of an external magnetic field.

This work is very innovative, has been well-conducted, the data is clearly presented and results are very convinced. This work has important implications in fields of magnetic response materials, supramolecular systems, smart materials and biomaterials, among others.

I recommend its publication as it is.

Reviewers' Comments:

Reviewer #1:

Remarks to the Author:

I noted that the author has performed additional experiments and revised their manuscript. While the revision partially addressed my concerns, I still find it unconvincing. This is because the magnetic assembly is vital, and the most reliable data to support it are AFM and TEM imaging. As mentioned earlier, AFM imaging in air cannot accurately illustrate the successful assembly, as the presence of salts and solutes in the solution during the drying process can significantly impact the morphology.

Reviewer #3:

Remarks to the Author:

The authors have now assigned the NMR spectrum to the Ala, Aib and PEG resonances. They could have used as additional argument for alpha-helicity of the peptide the H and N chemical shifts, but even without this CD data is convincing on its own.

I do not have more comments with respect to the NMR part of the work.

Reviewer #4:

Remarks to the Author:

The authors have addressed all the reviewers' queries, the article can now be published

Reviewer's comment and our response

Reviewer #1:

(Comment 1)

I noted that the author has performed additional experiments and revised their manuscript. While the revision partially addressed my concerns, I still find it unconvincing. This is because the magnetic assembly is vital, and the most reliable data to support it are AFM and TEM imaging. As mentioned earlier, AFM imaging in air cannot accurately illustrate the successful assembly, as the presence of salts and solutes in the solution during the drying process can significantly impact the morphology.

(Response 1)

Although the AFM and TEM images have been obtained in air, they can represent the most thermodynamically stable equilibrium morphologies. Moreover, we have verified and reproduced the results using a series of multiple compounds. In addition, the experiments performed in solution such as NMR and MCD (magnetic circular dichroism) confirmed that the designed molecules showed magnetic responsiveness in solution. Given the reviewer's comment, we added the following sentence in the Discussion section; "We have experimentally verified the magnetic responsiveness both in solution state and in thin films containing equilibrium morphologies"